# Latent Refinement Decoding: Enhancing Diffusion-Based Language Models by Refining Belief States

## Abstract

Autoregressive (AR) models remain the standard for natural language generation but still suffer from high latency due to strictly sequential decoding. Recent diffusion-inspired approaches, such as LlaDA and Dream, mitigate this by generating in parallel, yet they suffer from two core limitations: information loss, as predictive distributions for non-finalised tokens are discarded at each step, and a lack of well-behaved commitment dynamics, where local decisions are not properly coordinated at the global level. We introduce Latent Refinement Decoding (LRD), a two-stage framework with *Latent Refinement* and a *Predictive Feedback Loop*. The first stage maintains masked positions as distributional mixtures of predicted tokens and the mask embedding, allowing the model to establish more globally consistent beliefs. The second stage progressively finalises confident tokens while retaining uncertain ones for iterative feedback. KL-divergence dynamics provide a principled and reliable criterion for convergence and early stopping. Experiments across coding (HumanEval +6.3, MBPP +2.6) and reasoning (GSM8K +2.9, MATH500 +3.8) show that LRD improves accuracy while delivering speedups of up to 10.6×. Moreover, LRD is orthogonal to system-level optimisation: when combined with KV-cache and parallel based accelerators (e.g., Fast-dLLM), it improves accuracy and yields up to 2.4× additional speedup, making it a strong and versatile alternative for parallel sequence generation.

## 1 Introduction

Autoregressive (AR) models have long defined the standard for natural language generation (Brown et al., 2020; Fei et al., 2025; Achiam et al., 2023; Yang et al., 2025), but their inherently sequential token-by-token decoding imposes a fundamental bottleneck on inference latency (Touvron et al., 2023; Sun et al., 2024). This constraint has motivated the development of parallel decoding paradigms. Among them, diffusion-inspired approaches such as *LLaDA* (Nie et al., 2025) and *Dream* (Ye et al., 2025) offer a particularly promising direction. By formulating text generation as an iterative refinement process and updating all token positions in parallel at each step, these methods provide a compelling alternative to traditional AR decoding, achieving significant speedups while maintaining competitive quality (Labs et al., 2025; Deepmind, 2025). Despite recent progress, diffusion language models employ hard assignment strategies (Gong et al., 2025; Nie et al., 2025; Ye et al., 2025): at each denoising step, they commit high-confidence positions to specific tokens while resetting remaining positions to uniform `[MASK]` tokens. The predictive distributions from earlier steps are discarded, limiting the model's ability to build upon partial beliefs established in earlier iterations.

This design introduces two limitations: (i) **Information loss from hard masking** (Li et al., 2024): At each denoising step, positions below confidence thresholds are reset to uniform `[MASK]` embeddings, completely discarding their predictive distributions. This prevents uncertain positions from sharing probabilistic information through self-attention, forcing each masked position to be predicted in isolation. When mispredictions occur, the hard assignment yields infinite KL divergence from the true posterior, as it assigns zero probability mass to the correct token. (ii) **Lack of well-behaved commitment dynamics** (Luxembourg et al., 2025; Li & Cai, 2025): The binary nature of hard assignment creates a dilemma: **aggressive** selection commits early and can lock in incorrect predictions, propagating errors through later steps; **conservative** selection keeps many positions masked, which

slows progress and requires many denoising iterations. Moreover, using a fixed number of iterations ignores the varying complexity across different generation tasks, wasting computation on simple cases while potentially underserving complex ones.

To overcome these limitations, we move beyond purely discrete denoising and introduce **Latent Refinement Decoding (LRD)**, a hybrid framework that operates in both embedding and token spaces. LRD restructures the denoising process into two coordinated stages. **Phase 1: Latent Refinement** performs distribution-preserving updates entirely in the embedding space: for each masked position, we form a mix embedding by mixing the [MASK] embedding with the entropy-normalised expectation over top-$p$ predicted token embeddings, allowing the model to "think latently" in continuous embedding space, establishing globally coherent beliefs before committing to discrete decisions. Once the predictive distributions stabilise, **Phase 2: Predictive Feedback Loop** progressively converts low-entropy positions into discrete tokens while keeping the remaining positions in soft form, feeding each step's predictions back into the next soft mixture; KL-based monitors govern the soft-to-hard transition and enable adaptive early stopping. Specifically, **the main contributions of LRD are:**

1. **Soft diffusion** that enables continuous denoising in embedding space by mixing [MASK] with weighted token representations. This preserves distributional information across steps and enables cross-position refinement through self-attention.

2. **Adaptive two-phase sampling** that combines soft refinement for global coherence with hard decoding for precise convergence. KL-based monitoring enables automatic phase transitions and early stopping based on actual convergence rather than fixed iteration counts.

3. We validate LRD across diverse model families, generation lengths, and benchmarks spanning coding (HumanEval: +6.3, MBPP: +2.6) and reasoning (GSM8K: +2.9, MATH500: +3.8), and general language tasks (reading comprehension and summarisation), improving accuracy across all domains and achieving speedups of up to $10.6\times$.

4. LRD is orthogonal to system-level optimisations such as KV-cache and parallel decoding–based accelerators (e.g., Fast-dLLM). When combined with Fast-dLLM it improves accuracy and yields up to $2.4\times$ speedup, highlighting its value as a drop-in decoding module for future diffusion LMs.

## 2 PRELIMINARY

For dLLMs (Ou et al., 2024; Zheng et al., 2024; Shi et al., 2024; Gong et al., 2025), the forward process corrupts data $\mathbf{x}_0 \in \{1, ..., V\}^L$ (a sequence of $L$ tokens from vocabulary size $V$) into progressively noisier versions $\mathbf{x}_1, ..., \mathbf{x}_T$. At each timestep, the forward process is defined as a categorical distribution:

$$q(\mathbf{x}_t|\mathbf{x}_{t-1}) = \text{Cat}(\mathbf{x}_t; \mathbf{Q}_t^\top \mathbf{x}_{t-1}) \tag{1}$$

where $\mathbf{x}_t \in \{0, 1\}^{V \times L}$ is the one-hot representation of tokens at time $t$, and $\mathbf{Q}_t \in [0, 1]^{V \times V}$ is the transition matrix. Each token either remains unchanged with probability $1 - \beta_t$ or transitions to the special [MASK] token with probability $\beta_t \in (0, 1)$: $\mathbf{Q}_t = (1 - \beta_t)\mathbf{I} + \beta_t \mathbf{1}\mathbf{m}^\top$, where $\mathbf{I} \in \mathbb{R}^{V \times V}$ is the identity matrix, $\mathbf{1} \in \mathbb{R}^V$ is an all-ones vector, and $\mathbf{m} \in \{0, 1\}^V$ is the one-hot encoding of the [MASK] token. Under continuous-time formulation with $t \in [0, 1]$, the cumulative transition matrix from $\mathbf{x}_0$ to $\mathbf{x}_t$ becomes: $\overline{\mathbf{Q}}_t = \alpha_t^* \mathbf{I} + (1 - \alpha_t^*)\mathbf{1}\mathbf{m}^\top$, where $\alpha_t^* = \prod_{s=1}^t (1 - \beta_s)$ represents the probability of a token remaining unmasked from time 0 to time $t$.

The reverse process $p_\theta(\mathbf{x}_{t-1}|\mathbf{x}_t)$ aims to reconstruct the original data by iteratively denoising from $\mathbf{x}_T$ (fully masked) to $\mathbf{x}_0$ (clean text). At each denoising step $t$, the model predicts a distribution over tokens for each position: $\hat{p}_\theta^{(i)}(\mathbf{x}_0|\mathbf{x}_t)$ for position $i$.

In transformer-based diffusion models, each token is represented by a learnable embedding vector. Let $\mathbf{e}_v \in \mathbb{R}^d$ denote the embedding for token $v \in V$, and $\mathbf{e}_{[\text{MASK}]} \in \mathbb{R}^d$ the embedding for the [MASK] token. During the reverse process, traditional sampling strategies employ **hard assignment**, selecting tokens based on prediction confidence:

$$v_t^{(i)} = \begin{cases} \arg\max_{v \in V} \hat{p}_\theta^{(i)}(v|\mathbf{x}_t), & \text{if } i \in \text{top-1}(\{H_t^{(j)}\}_{j=1}^L) \\ [\text{MASK}], & \text{otherwise} \end{cases} \tag{2}$$

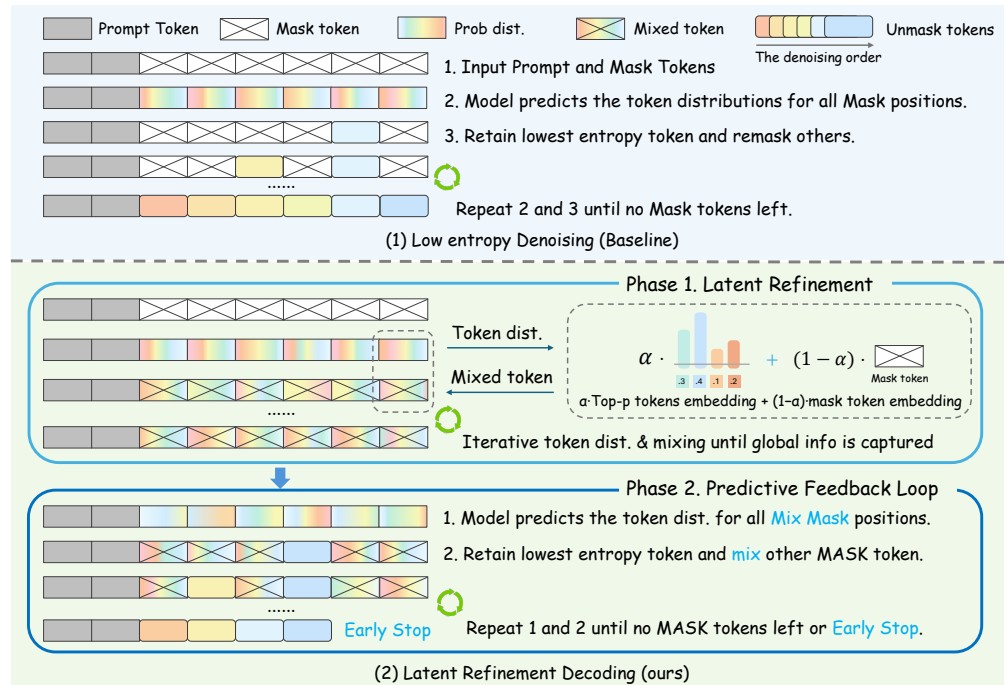

Figure 1: Comparison between the existing decoding strategy and the proposed method. Different colours represent distinct tokens, while gradient colours indicate predicted token representations. **Top:** In the existing strategy, all [MASK] tokens share the same embedding and are repeatedly remasked if not selected. **Bottom:** In LRD, Phase 1 refines each [MASK] embedding, and Phase 2 progressively commits confident tokens while keeping uncertain ones soft for context-aware decoding.

where $H_t^{(j)} = -\sum_v \hat{p}_\theta^{(j)}(v|\mathbf{x}_t) \log \hat{p}_\theta^{(j)}(v|\mathbf{x}_t)$ is the entropy at position $j$, and top-1 selects the position with lowest entropy (highest confidence). This creates a binary embedding assignment: each position uses either $\mathbf{e}_{[\text{MASK}]}$ or a specific token embedding $\mathbf{e}_{v_t^{(i)}}$, resulting in complete information loss for positions not selected. This binary decision mechanism creates a discontinuous mapping from probability distributions to discrete embeddings: positions below the confidence threshold are reset to pure [MASK] embeddings, completely discarding their distributional information $\hat{p}_\theta(\cdot|\mathbf{x}_t)$, resulting in abrupt information loss and suboptimal exploration of the posterior distribution.

## 3 METHODOLOGY

Effective discrete diffusion sampling requires maintaining sufficient uncertainty for exploration while gradually incorporating token-specific information for convergence. To achieve this balance, we propose LRD: instead of binary decisions that abruptly switch between pure noise ([MASK]) and deterministic tokens, we create intermediate representations through continuous embedding interpolation. Specifically, we construct mixed embeddings that blend [MASK] and token embeddings weighted by prediction uncertainty, where high-entropy positions retain more mask-like characteristics (preserving exploration) while low-entropy positions incorporate more token information (enabling commitment). This enables a gradual denoising trajectory where the noise-signal ratio smoothly decreases, yielding better-calibrated probability distributions for subsequent sampling steps.

### 3.1 SOFT DIFFUSION

Our method operates in the embedding space rather than discrete token space. At each timestep $t$, we maintain a set of *soft embeddings* $\mathcal{E}_t = \{\tilde{\mathbf{e}}_t^{(1)}, \ldots, \tilde{\mathbf{e}}_t^{(N)}\}$ where

$$\tilde{\mathbf{e}}_t^{(i)} = (1 - \alpha_t^{(i)}) \cdot \mathbf{e}_{[\text{MASK}]} + \alpha_t^{(i)} \cdot \sum_{v \in \mathcal{T}_{t+1}^{(i)}} \bar{p}_{t+1}^{(i)}(v) \cdot \mathbf{e}_v \quad (3)$$

Here, $\mathbf{e}_v \in \mathbb{R}^d$ denotes the embedding of the token $v \in \mathcal{T}_{t+1}^{(i)}$, where $\mathcal{T}_{t+1}^{(i)}$ is the top-$p$ nucleus set. $\mathbf{e}_{\texttt{[MASK]}}$ is the $\texttt{[MASK]}$ embedding, $\bar{p}_{t+1}^{(i)}(v)$ denotes the probability mass of token $v$ at position $i$, renormalised to the nucleus set $\mathcal{T}_{t+1}^{(i)}$. The coefficient $\alpha_t^{(i)} \in [0, 1]$ controls the interpolation strength. The mixing weight $\alpha_t$ is controlled by entropy:

$$\alpha_t^{(i)} = r_f \cdot (1 - \hat{H}_{t+1}^{(i)}) = r_f \cdot (1 + \frac{\sum_{k=1}^{|V|} p_{t+1}^{(i)}(k) \log p_{t+1}^{(i)}(k)}{\log |V|}) \tag{4}$$

where $p_{t+1}^{(i)}(k)$ refers to the probability distribution over the full vocabulary, $\hat{H}_{t+1}^{(i)}$ is the normalised entropy of this distribution, and $r_f \in (0, 1]$ sets the maximum interpolation strength. Since the entropy of a categorical distribution over a vocabulary of size $V$ lies in $[0, \log |V|]$, we divide by $\log |V|$ to normalise it into $[0, 1]$. This design ensures that uncertain positions stay mask-like while confident ones commit to tokens. Formal justification and stability analysis are deferred to Appendix D.

Consider the absorbing discrete diffusion process where the true posterior distribution $q^*(x_{t-1}|x_t, x_0)$ represents optimal denoising. For masked positions where $x_t^{(i)} = \texttt{[MASK]}$, Bayes' rule yields:

$$q^*(x_{t-1}^{(i)}|x_t^{(i)} = \texttt{[MASK]}, x_0^{(i)}) = \frac{\alpha_{t-1}^* - \alpha_t^*}{1 - \alpha_t^*} \delta_{x_0^{(i)}} + \frac{1 - \alpha_{t-1}^*}{1 - \alpha_t^*} \delta_{\texttt{[MASK]}} \tag{5}$$

where the detailed derivation is provided in Appendix B. Hard assignment approximates this by a degenerate distribution $\hat{q}_{\text{hard}} \in \{\delta_{\hat{x}_0^{(i)}}, \delta_{\texttt{[MASK]}}\}$. If $\hat{x}_0^{(i)} \neq x_0^{(i)}$, then $\hat{q}_{\text{hard}}$ assigns zero probability where $q^*$ is positive, leading to $\text{KL}(q^* \| \hat{q}_{\text{hard}}) = \infty$. Moreover, positions that remain masked are represented by a fixed embedding $\mathbf{e}_{\texttt{[MASK]}}$, which conveys no distributional information to neighbouring positions.

Latent Refinement Decoding mitigates both issues. First, $\hat{q}_{\text{soft}}$ assigns non-zero probability to all tokens, ensuring the true token retains a positive mass even under misprediction. Second, the weighted mixture $\sum_v \bar{p}_t^{(i)}(v) \mathbf{e}_v$ can be viewed as the expected embedding under the model's belief at position $i$. Since self-attention is linear in the embeddings, this representation propagates uncertainty information across positions, enabling different tokens to condition on each other's belief states.

## 3.2 Adaptive Sampling with Soft-to-Hard Scheduling

The optimal denoising strategy must balance two objectives: preserving sufficient uncertainty for exploration while progressively reducing entropy for convergence. Latent Refinement Decoding provides a smooth relaxation in the embedding space, where gradient-based updates are well behaved and guarantee contraction toward fixed points. This geometry enables rapid early progress, as the gradients carry informative signals across the entire vocabulary. However, Latent Refinement Decoding cannot fully collapse distributions to one-hot states, since embeddings always encode mixtures rather than discrete commitments. As a result, convergence slows in later stages when sharper updates are required for final token generation.

To overcome this limitation, we adopt a two-phase schedule. **Phase 1** exploits the favourable geometry of soft embeddings to quickly reach a stable neighborhood of the optimum. Once the model's predictive distributions stabilise ($\mathcal{D}_{\text{KL}}^{(t)} < \tau_{\text{refine}}$), **Phase 2** transitions to hard assignment, which enables decisive discrete optimisation within the well-conditioned basin. This design follows the principle of graduated optimisation: begin with a smooth relaxation to encourage global exploration, then progressively sharpen the objective to encourage convergence.

**Phase 1: Latent Refinement via Soft Embeddings.** During the initial refinement phase, the model iteratively refines predictive distributions through soft embedding propagation without committing to any discrete tokens. Starting from $t = T$ (fully masked), we compute soft embeddings using Equation 3, where predictions $p_t^{(i)}(v) = dLLM_\theta(\tilde{\mathcal{E}}_t)^{(i)}$ are conditioned on the previous soft embeddings rather than discrete tokens. This allows distributional information to propagate across timesteps.

As refinement progresses, the soft embeddings approach a fixed point where the model's predictions become self-consistent, that is, the output distribution given the current soft embeddings closely matches the distribution encoded in those embeddings. At this convergence point, the model has

extracted all available information from the global distributional structure and further soft refinement yields diminishing returns. We detect this saturation by monitoring the KL divergence between consecutive predictions:

$$\mathcal{D}_{\text{KL}}^{(t)} = \frac{1}{L} \sum_{i=1}^{L} D_{\text{KL}}(p_t^{(i)} \| p_{t+1}^{(i)}) \tag{6}$$

When $\mathcal{D}_{\text{KL}}^{(t)} < \tau_{\text{refine}}$, the belief state has stabilised, indicating that the model can no longer benefit from the soft embedding's global information and requires discrete commitments to make further progress. This triggers the transition to Phase 2, where discrete token generation can exploit the well-initialised distributions from Phase 1. Alternatively, if convergence is not achieved within $T_{\text{refine}}$ steps, we still transition to Phase 2 for computational efficiency, as extended refinement shows diminishing returns while incurring additional computational cost.

**Phase 2: Predictive Feedback Loop.** Once convergence is detected at timestep $t^*$, we switch to Predictive Feedback decoding for the remaining timesteps $t \in [t^*, 0]$. We modify the standard hard assignment (Equation 2) by replacing `[MASK]` embeddings with soft embeddings for unselected positions:

$$\mathbf{e}_t^{(i)} = \begin{cases} \mathbf{e}_{\arg\max_v p_t^{(i)}(v)}, & \text{if } i \in \text{top-1}(\{H_t^{(j)}\}_{j=1}^L) \\ \tilde{\mathbf{e}}_t^{(i)}, & \text{otherwise} \end{cases} \tag{7}$$

This preserves the distributional information from Phase 1's refinement in uncommitted positions, providing richer context for subsequent decoding steps while still allowing confident positions to make discrete commitments.

During decoding, we continue monitoring $\mathcal{D}_{\text{KL}}^{(t)}$ (Equation 6). If $\mathcal{D}_{\text{KL}}^{(t)} < \tau_{\text{decode}}$, the predictive distributions over the whole sentence have converged to a stable configuration and further iterations would be redundant. This early stopping mechanism terminates the generation and outputs the final sequence, ensuring computational efficiency without sacrificing output quality. In practice, this allows the model to adaptively adjust its generation length based on the problem complexity rather than using a fixed number of steps.

## 4 EXPERIMENTS

### 4.1 IMPLEMENTATION DETAILS

We evaluate our method on two representative diffusion-based language models: LLaDA 8B (Nie et al., 2025; Zhu et al., 2025) and Dream 7B (Ye et al., 2025), each with both Base and Instruct variants. To ensure robustness, we fix the temperature to 0 and always select the token with the minimum entropy at each decoding step, detailed configuration in Appendix A.1. All experiments are conducted on a server equipped with 8 NVIDIA A100 80GB GPUs.

### 4.2 BENCHMARKS AND METRICS

To comprehensively assess the effectiveness of our approach, we conduct experiments on four benchmarks spanning mathematical reasoning and code generation. For mathematical reasoning, we use GSM8K (Cobbe et al., 2021), which consists of grade-school math word problems, and the more challenging MATH500 (Lightman et al., 2024), a benchmark of competition-level mathematics problems. For code generation, we evaluate on MBPP (Austin et al., 2021b), which features entry-level Python programming tasks, and HumanEval (Chen et al., 2021), a set of handwritten coding problems for program synthesis. Following prior work, all Instruct models are evaluated under the zero-shot setting. For Base models, we follow standard few-shot settings for each benchmark: zero-shot for HumanEval, 3-shot for MBPP, 4-shot for MATH500, and 8-shot for GSM8K. For all benchmarks, we report accuracy for mathematical reasoning and pass@1 for code generation.

### 4.3 MAIN RESULTS

**Performance on Benchmarks.** Table 1 reports the performance of different models and decoding methods across four representative benchmarks. Our Latent Refinement Decoding framework

Table 1: Performance of different models and methods across benchmarks. Speed denotes relative runtime (baseline = 1.0×), where larger values indicate faster and more efficient inference. Baseline results are shown in grey, and ours LRD improvements in green.

| Model | Len | Method | HumanEval | | MBPP | | GSM8K | | MATH500 | |
|---|---|---|---|---|---|---|---|---|---|---|
| | | | Acc | Speed | Acc | Speed | Acc | Speed | Acc | Speed |
| **Dream-Base-7B** | 256 | baseline | 50.6 | 1.0× | 55.8 | 1.0× | 75.3 | 1.0× | 36.9 | 1.0× |
| | | Ours | $56.9_{+6.3}$ | 1.2× | $57.6_{+1.8}$ | 2.3× | $78.2_{+2.9}$ | 1.8× | $39.8_{+2.9}$ | 1.4× |
| | 512 | baseline | 54.4 | 1.0× | 55.8 | 1.0× | 76.2 | 1.0× | 37.5 | 1.0× |
| | | Ours | $58.8_{+4.4}$ | 2.6× | $58.4_{+2.6}$ | 4.5× | $77.4_{+1.2}$ | 3.4× | $40.8_{+3.3}$ | 1.8× |
| | 1024 | baseline | 54.8 | 1.0× | 58.0 | 1.0× | 76.8 | 1.0× | 39.1 | 1.0× |
| | | Ours | $59.1_{+4.3}$ | 4.4× | $58.8_{+0.8}$ | 7.6× | $77.8_{+1.0}$ | 4.2× | $42.4_{+3.3}$ | 2.2× |
| **Dream-Ins-7B** | 256 | baseline | 55.4 | 1.0× | 57.4 | 1.0× | 80.8 | 1.0× | 37.9 | 1.0× |
| | | Ours | $61.6_{+6.2}$ | 1.4× | $59.4_{+2.0}$ | 2.4× | $83.0_{+2.2}$ | 1.4× | $40.6_{+2.7}$ | 1.1× |
| | 512 | baseline | 56.1 | 1.0× | 56.7 | 1.0× | 80.2 | 1.0× | 38.6 | 1.0× |
| | | Ours | $60.9_{+4.8}$ | 2.9× | $58.8_{+2.1}$ | 4.6× | $82.7_{+2.5}$ | 3.6× | $41.8_{+3.2}$ | 1.2× |
| | 1024 | baseline | 56.0 | 1.0× | 57.3 | 1.0× | 81.3 | 1.0× | 40.1 | 1.0× |
| | | Ours | $61.0_{+5.0}$ | 9.3× | $59.0_{+1.7}$ | 10.6× | $83.5_{+2.2}$ | 5.5× | $43.9_{+3.8}$ | 1.7× |
| **LLaDA-Base-8B** | 256 | baseline | 32.9 | 1.0× | 39.7 | 1.0× | 69.1 | 1.0× | 30.2 | 1.0× |
| | | Ours | $36.0_{+3.1}$ | 1.3× | $41.4_{+1.7}$ | 1.5× | $71.2_{+2.1}$ | 1.6× | $32.4_{+2.2}$ | 1.4× |
| | 512 | baseline | 32.8 | 1.0× | 39.8 | 1.0× | 70.8 | 1.0× | 30.8 | 1.0× |
| | | Ours | $36.0_{+3.2}$ | 1.7× | $41.4_{+1.6}$ | 1.9× | $72.5_{+1.7}$ | 2.2× | $32.4_{+1.6}$ | 1.6× |
| | 1024 | baseline | 31.7 | 1.0× | 39.8 | 1.0× | 71.4 | 1.0× | 30.1 | 1.0× |
| | | Ours | $34.8_{+3.1}$ | 2.2× | $40.8_{+1.0}$ | 3.6× | $72.1_{+0.7}$ | 3.3× | $32.2_{+2.1}$ | 2.1× |
| **LLaDA-Ins-8B** | 256 | baseline | 38.7 | 1.0× | 36.9 | 1.0× | 77.4 | 1.0× | 33.8 | 1.0× |
| | | Ours | $43.3_{+4.6}$ | 1.2× | $40.0_{+3.1}$ | 1.3× | $78.8_{+1.4}$ | 1.5× | $35.8_{+2.0}$ | 1.4× |
| | 512 | baseline | 43.9 | 1.0× | 38.2 | 1.0× | 81.3 | 1.0× | 37.7 | 1.0× |
| | | Ours | $48.4_{+4.5}$ | 1.3× | $40.6_{+2.4}$ | 1.5× | $84.5_{+3.2}$ | 2.0× | $39.8_{+2.1}$ | 1.4× |
| | 1024 | baseline | 44.6 | 1.0× | 37.4 | 1.0× | 82.3 | 1.0× | 39.4 | 1.0× |
| | | Ours | $49.5_{+4.9}$ | 1.7× | $39.6_{+2.2}$ | 3.7× | $83.7_{+1.4}$ | 4.3× | $42.2_{+2.8}$ | 2.0× |
| **LLaDA-1.5-8B** | 256 | baseline | 38.4 | 1.0× | 38.6 | 1.0× | 79.2 | 1.0× | 33.4 | 1.0× |
| | | Ours | $44.5_{+6.1}$ | 1.2× | $39.8_{+1.2}$ | 1.3× | $80.4_{+1.2}$ | 1.5× | $36.6_{+3.2}$ | 1.3× |
| | 512 | baseline | 45.1 | 1.0× | 37.6 | 1.0× | 82.9 | 1.0× | 38.6 | 1.0× |
| | | Ours | $49.6_{+4.5}$ | 1.2× | $40.2_{+2.6}$ | 1.5× | $84.5_{+1.6}$ | 1.9× | $41.0_{+2.4}$ | 1.4× |
| | 1024 | baseline | 45.7 | 1.0× | 37.4 | 1.0× | 82.5 | 1.0× | 39.6 | 1.0× |
| | | Ours | $50.6_{+4.9}$ | 1.7× | $39.6_{+2.2}$ | 3.5× | $83.9_{+1.4}$ | 4.0× | $41.8_{+2.2}$ | 1.9× |

consistently improves accuracy across all settings. For instance, on HumanEval, LRD boosts pass@1 by up to +6.3 points (Dream-Base-7B, 256 tokens) and +6.2 points (Dream-Ins-7B, 256 tokens) compared to the baseline. Similar trends are observed for MBPP, GSM8K, and MATH500, where our method outperforms the baseline by margins of +1.0 to +4.8 points in most cases. These results are consistent across different sequence lengths (256, 512, 1024), confirming that the benefits of LRD are robust to context window size and apply uniformly to both Base and Instruct model families.

**Efficiency and Decoding Speed.** Beyond accuracy, LRD substantially accelerates inference. As shown in Table 1, our method delivers at least $1.2\times$ speedup in all cases, with the largest gains observed for longer contexts. For example, Dream-Ins-7B achieves up to $9.3\times$ faster decoding at length 1024, while LLaDA models reach up to $4.3\times$ speedup under the same condition. The improvement comes from two factors: (i) the mix operation in the latent refinement phase accelerates convergence by reducing the number of tokens that need to be generated (see Section 4.5), and (ii) the entropy-based early stopping criterion prevents unnecessary refinement steps, especially in long sequences. These results indicate that LRD is particularly advantageous in large-context scenarios, where traditional parallel decoding incurs significant overhead.

## 4.4 CONVERGENCE ANALYSIS

**KL divergence decreases steadily during refinement and decoding.** Figure 2 shows the KL divergence between step-wise predictive distributions and the final decoded outputs for LLaDA-1.5 and Dream-Ins across four benchmarks. For ease of observation, we fix the latent refinement phase to 20 steps. The divergence exhibits a clear downward trend: during the latent refinement phase, the KL values drop rapidly and stabilise, indicating that the latent belief state quickly converges before decoding begins. Once decoding starts, the KL divergence continues to decrease with mild fluctuations, reflecting the model's progressive confidence sharpening. For most benchmarks, the divergence approaches zero within about 300 steps, whereas Dream-Ins converges even faster, reaching near-zero divergence around 140 steps. The MATH500 benchmark proves more challenging,

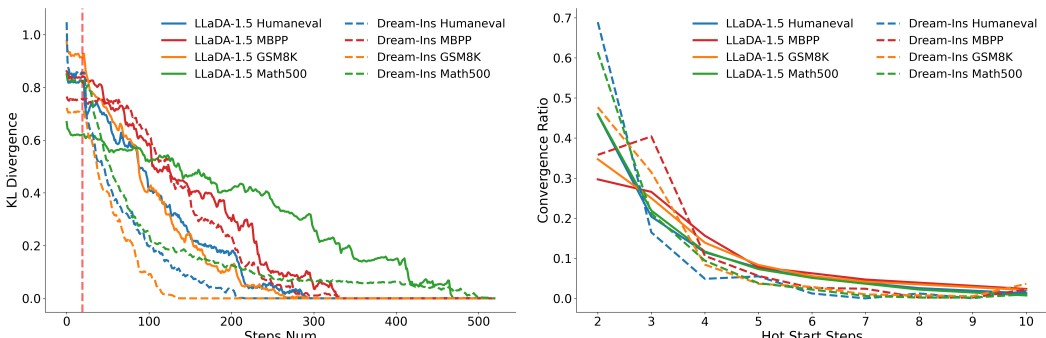

Figure 2: KL divergence between step-wise predictive distributions and final decoded results for LLaDA-1.5 and Dream-Ins across benchmarks. The red vertical line marks where decoding begins after a fixed 20-step latent refinement.

Figure 3: Convergence ratios across latent refinement steps for LLaDA-1.5 and Dream-Ins on four benchmarks. Since computing the difference in KL divergence requires at least three consecutive steps, the curves are plotted starting from step 2.

with non-negligible divergence persisting until the full 512-step horizon. Overall, these patterns are consistent with our expectations: the refinement phase provides a stable initialisation, and the subsequent decoding stage steadily drives the system toward convergence.

**Most examples converge within the first few latent refinement steps.** Figure 3 reports the proportion of cases converging at each latent refinement step. Across benchmarks, most runs converge within the first few refinement steps. On HumanEval with Dream-Ins, 68.9% of samples converge by step 2, and more than 85% by step 3. Similar trends hold for GSM8K, MBPP, and MATH500, where over 70% of cases converge within the first three to four steps. These results confirm that latent refinement is highly efficient in practice: most examples stabilize early, reducing the need for excessive refinement iterations and validating the design of our latent refinement mechanism.

### 4.5 ABLATION STUDY

Table 2: Ablation study on decoding variants, reporting Speed and effective token number $E_{\text{token}}$, where red and green numbers show the change compared to our full method.

| Length | Method | Speed | | | | $E_{\text{token}}$ | | | |
|---|---|---|---|---|---|---|---|---|---|
| | | HumanEval | MBPP | GSM8K | MATH500 | HumanEval | MBPP | GSM8K | MATH500 |
| 256 | baseline | 1.0× | 1.0× | 1.0× | 1.0× | 117.2 | 53.5 | 132.4 | 228.4 |
| | Ours | $1.4\times_{+0.4}$ | $2.4\times_{+1.4}$ | $1.4\times_{+0.4}$ | $1.1\times_{+0.1}$ | $108.4_{-8.8}$ | $49.2_{-4.3}$ | $128.6_{-5.8}$ | $226.0_{-2.4}$ |
| | w/o latent refinement | $1.5\times_{+0.1}$ | $2.5\times_{+0.1}$ | $1.5\times_{+0.1}$ | $1.1\times_{+0.0}$ | $108.7_{+0.3}$ | $50.4_{+1.2}$ | $129.9_{+1.3}$ | $226.8_{+0.8}$ |
| | w/o mix embed | $1.3\times_{-0.1}$ | $2.2\times_{-0.2}$ | $1.5\times_{+0.1}$ | $1.0\times_{-0.1}$ | $117.2_{+8.8}$ | $49.5_{+0.3}$ | $129.4_{+0.8}$ | $228.4_{+2.4}$ |
| | w/o early stop | $0.8\times_{-0.6}$ | $0.7\times_{-1.7}$ | $0.8\times_{-0.6}$ | $0.9\times_{-0.2}$ | $109.7_{+1.3}$ | $51.4_{+2.2}$ | $129.9_{+1.3}$ | $228.0_{+2.0}$ |
| 512 | baseline | 1.0× | 1.0× | 1.0× | 1.0× | 116.2 | 55.7 | 135.2 | 378.9 |
| | Ours | $2.9\times_{+1.9}$ | $4.6\times_{+3.6}$ | $3.6\times_{+2.6}$ | $1.2\times_{+0.2}$ | $103.9_{-12.3}$ | $51.8_{-4.1}$ | $125.9_{-9.3}$ | $363.5_{-15.4}$ |
| | w/o latent refinement | $3.1\times_{+0.2}$ | $4.9\times_{+0.3}$ | $3.8\times_{+0.2}$ | $1.2\times_{+0.0}$ | $106.3_{+2.4}$ | $52.6_{+0.8}$ | $127.9_{+2.0}$ | $363.0_{-0.5}$ |
| | w/o mix embed | $2.7\times_{-0.2}$ | $4.3\times_{-0.3}$ | $3.0\times_{-0.6}$ | $1.0\times_{-0.2}$ | $116.2_{+12.3}$ | $51.8_{+0.0}$ | $126.2_{+0.3}$ | $368.9_{+5.4}$ |
| | w/o early stop | $0.8\times_{-2.1}$ | $0.7\times_{-3.9}$ | $0.8\times_{-2.8}$ | $0.8\times_{-0.4}$ | $106.2_{+2.3}$ | $53.6_{+1.8}$ | $127.2_{+1.3}$ | $366.0_{+2.5}$ |
| 1024 | baseline | 1.0× | 1.0× | 1.0× | 1.0× | 90.4 | 60.5 | 135.5 | 482.3 |
| | Ours | $9.3\times_{+8.3}$ | $10.6\times_{+9.6}$ | $5.5\times_{+4.5}$ | $1.7\times_{+0.7}$ | $84.6_{-5.8}$ | $57.2_{-3.3}$ | $123.7_{-11.8}$ | $437.3_{-45.0}$ |
| | w/o latent refinement | $9.3\times_{+0.0}$ | $10.7\times_{+0.1}$ | $5.6\times_{+0.1}$ | $1.7\times_{+0.0}$ | $83.9_{-0.7}$ | $58.2_{+1.0}$ | $125.0_{+1.3}$ | $455.4_{+18.1}$ |
| | w/o mix embed | $9.1\times_{-0.2}$ | $10.4\times_{-0.2}$ | $5.1\times_{-0.4}$ | $1.3\times_{-0.4}$ | $90.4_{+5.8}$ | $61.7_{+4.5}$ | $130.5_{+6.8}$ | $483.5_{+46.2}$ |
| | w/o early stop | $0.8\times_{-8.5}$ | $0.7\times_{-9.9}$ | $0.8\times_{-4.7}$ | $0.8\times_{-0.9}$ | $86.2_{+1.6}$ | $59.2_{+2.0}$ | $126.1_{+2.4}$ | $438.9_{+1.6}$ |

**Latent refinement slows generation, mixed embeddings aid convergence, and early stopping is the main accelerator.** Table 2 reveals several key insights. First, removing the latent refinement phase (w/o latent refinement) yields faster decoding, showing that latent refinement introduces extra refinement steps and slightly slows down speed, though it improves stability. Second, removing mixed embeddings (w/o mix embed) makes decoding slower and increases effective token counts, indicating that mixing embeddings is critical for helping the model converge earlier. Third, early stopping (w/o early stop) leads to dramatic slowdowns, with speed dropping from multi-fold acceleration to even below baseline, despite only negligible changes in $E_{\text{token}}$. This confirms that early stopping is the primary driver of speedup. Finally, both latent refinement and mixed embeddings reduce effective

token usage under the full model, demonstrating that they improve convergence efficiency even though their speed impact differs.

**Both components contribute, with mixing more critical.** Table 3 reports ablation results in accuracy. Removing latent refinement or mixed embeddings consistently reduces performance, confirming the importance of both. The absence of mixed embeddings causes larger drops (up to $-2.9$ on HumanEval and $-2.5$ on MATH500), showing that the predictive feedback loop is the key driver of improvements. In contrast, early stopping incurs almost no accuracy loss while providing substantial efficiency gains. Overall, latent refinement and mixed embeddings are essential for accuracy, whereas early stopping boosts efficiency at virtually no cost.

**Excessive latent refinement brings no benefit but slows decoding.** Table 4 compares our two-stage strategy (one initial latent refinement followed by standard decoding) with variants that enforce latent refinement at every step, either a fixed number of times ($LF \times k$) or adaptively (Auto). Results show that while all variants outperform the baseline, none surpass our method: enforcing repeated latent refinements ($LF \times 2$–$5$) generally degrades accuracy, and even adaptive scheduling (Auto) underperforms compared to ours. The reason is that excessive latent refinement adds redundant computation without pro-

Table 3: Ablation study on decoding variants, where red and green numbers show the change compared to our full method.

| Len | Method | HumanEval | MBPP | GSM8K | MATH500 |
|---|---|---|---|---|---|
| | baseline | 55.4 | 57.4 | 80.8 | 37.9 |
| | Ours | $61.6_{+6.2}$ | $59.4_{+2.0}$ | $83.0_{+2.2}$ | $40.6_{+2.7}$ |
| 256 | w/o latent refinement | $60.1_{-1.5}$ | $58.6_{-0.8}$ | $82.3_{-0.7}$ | $39.4_{-1.2}$ |
| | w/o mix embed | $59.5_{-2.1}$ | $58.8_{-0.6}$ | $82.7_{-0.3}$ | $38.9_{-1.7}$ |
| | w/o early stop | $61.8_{+0.2}$ | $59.4_{+0.0}$ | $83.2_{+0.2}$ | $40.6_{+0.0}$ |
| | baseline | 56.1 | 56.7 | 80.2 | 38.6 |
| | Ours | $60.9_{+4.8}$ | $58.8_{+2.1}$ | $82.7_{+2.5}$ | $41.8_{+3.2}$ |
| 512 | w/o latent refinement | $59.9_{-1.0}$ | $57.8_{-1.0}$ | $82.2_{-0.5}$ | $41.0_{-0.8}$ |
| | w/o mix embed | $58.0_{-2.9}$ | $57.8_{-1.0}$ | $80.7_{-2.0}$ | $40.8_{-1.0}$ |
| | w/o early stop | $61.2_{+0.3}$ | $58.8_{+0.0}$ | $82.9_{+0.2}$ | $41.9_{+0.1}$ |
| | baseline | 56.0 | 57.3 | 81.3 | 40.1 |
| | Ours | $61.0_{+5.0}$ | $59.0_{+1.7}$ | $83.5_{+2.2}$ | $43.9_{+3.8}$ |
| 1024 | w/o latent refinement | $60.7_{-0.3}$ | $58.8_{-0.2}$ | $83.2_{-0.3}$ | $42.4_{-1.5}$ |
| | w/o mix embed | $59.1_{-1.9}$ | $58.7_{-0.3}$ | $82.9_{-0.6}$ | $41.4_{-2.5}$ |
| | w/o early stop | $61.4_{+0.4}$ | $59.0_{+0.0}$ | $83.7_{+0.2}$ | $44.2_{+0.3}$ |

Table 4: Ablation study on decoding variants at length 512, where *Auto* uses adaptive latent refinement, and *LF×k* enforces $k$ latent refinement steps prior to each token commitment.

| Method | HumanEval | MBPP | GSM8K | MATH500 |
|---|---|---|---|---|
| Baseline | 56.1 | 56.7 | 80.2 | 38.6 |
| Ours | $60.9_{+4.8}$ | $58.8_{+2.1}$ | $82.7_{+2.5}$ | $41.8_{+3.2}$ |
| Auto | $59.6_{+3.5}$ | $57.7_{+1.0}$ | $81.5_{+1.3}$ | $41.4_{+2.8}$ |
| LF×1 | $60.4_{+4.3}$ | $57.8_{+1.1}$ | $81.6_{+1.4}$ | $40.2_{+1.6}$ |
| LF×2 | $58.3_{+2.2}$ | $57.2_{+0.5}$ | $81.2_{+1.0}$ | $40.2_{+1.6}$ |
| LF×3 | $57.9_{+1.8}$ | $57.8_{+1.1}$ | $81.8_{+1.6}$ | $39.0_{+0.4}$ |
| LF×4 | $60.8_{+4.7}$ | $57.2_{+0.5}$ | $80.9_{+0.7}$ | $39.6_{+1.0}$ |
| LF×5 | $58.8_{+2.7}$ | $57.6_{+0.9}$ | $80.7_{+0.5}$ | $39.2_{+0.6}$ |

viding additional guidance once the model has stabilised. In contrast, our two-stage design strikes a better balance by leveraging latent refinement only at the beginning, yielding both higher accuracy and substantially faster decoding.

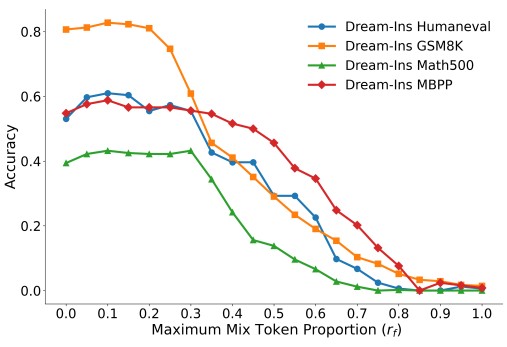

Figure 4: Accuracy of Dream-Ins on four benchmarks under different Maximum token proportion, where $r_f$=0 corresponds to the no mixing.

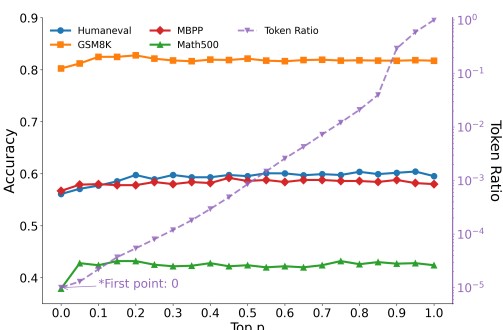

Figure 5: Effect of top-$p$ mixing on Dream-Ins across four benchmarks. The purple curve shows the log fraction of tokens included in the mixture.

**Full mixing collapses the model, while best at intermediate $r_f$.** We further investigate the effect of the maximum mix ratio $r_f$, which scales the interpolation between predicted token embeddings and the [MASK] embedding during refinement (Eq. 4). When $r_f$=0, the model falls back to always using the [MASK] token for unfinalised positions, equivalent to the baseline. At the other extreme, setting $r_f$=1 allows the mixing weight to fully follow the entropy schedule, meaning that in high-entropy cases the [MASK] embedding may vanish. As shown in Figure 4, both extremes are suboptimal: the baseline propagates information slowly, while overly aggressive mixing destabilises refinement

and leads to collapse. Intermediate values of $r_f$ achieve the best trade-off, providing sufficient mask guidance while still leveraging predictive feedback.

**Mix matters more than how many tokens are mixed.** As shown in Figure 5, when $p = 0$ no mixing occurs and the method degenerates to the baseline, giving the lowest accuracy across all benchmarks. Increasing $p$ quickly improves performance, even though the token ratio curve indicates that only a very small fraction of tokens are mixed at $p \leq 0.2$. This suggests that the key factor is enabling mixing rather than the absolute number of tokens included. Beyond $p \approx 0.2$, accuracy stabilises and fluctuates slightly, showing that adding more low-probability tokens offers little benefit while introducing potential noise. These results confirm that top-$p$ mixing provides a good balance: minimal mixing is already highly effective, and larger $p$ values do not bring further gains.

Table 5: Evaluation of Our Method Integrated with Fast-dLLM on Humaneval and GSM8K. Baseline results are shown in grey, improvements from our method are highlighted in green, and the orange indicate the additional speedup relative to the corresponding Fast-dLLM variant.

| Model | Method | HE Acc | HE Speed | GSM8K Acc | GSM8K Speed |
|---|---|---|---|---|---|
| **Dream-base-7B** | Baseline | 54.3 | 1.0× | 76.0 | 1.0× |
| | Fast-dLLM(prefix cache+parallel) | 54.2$_{-0.1}$ | 2.4× | 74.5$_{-1.5}$ | 4.8× |
| | Fast-dLLM(prefix cache+parallel)+ours | 56.0$_{+1.7}$ | 5.7×$_{2.4×}$ | 77.6$_{+1.6}$ | 13.1×$_{2.7×}$ |
| | Fast-dLLM(dual cache+parallel) | 54.2$_{-0.1}$ | 3.0× | 74.2$_{-1.8}$ | 6.0× |
| | Fast-dLLM(dual cache+parallel)+ours | 56.3$_{+2.0}$ | 6.7×$_{2.2×}$ | 77.3$_{+1.3}$ | 15.2×$_{2.5×}$ |
| **LLaDA-Ins-8B** | Baseline | 43.5 | 1.0× | 77.1 | 1.0× |
| | Fast-dLLM(prefix cache+parallel) | 43.4$_{-0.1}$ | 2.8× | 76.8$_{-0.3}$ | 10.4× |
| | Fast-dLLM(prefix cache+parallel)+ours | 45.8$_{+2.3}$ | 3.8×$_{1.4×}$ | 78.8$_{+1.7}$ | 11.8×$_{1.1×}$ |
| | Fast-dLLM(dual cache+parallel) | 43.2$_{-0.3}$ | 3.2× | 76.7$_{-0.4}$ | 11.3× |
| | Fast-dLLM(dual cache+parallel)+ours | 45.7$_{+2.2}$ | 4.7×$_{1.5×}$ | 78.7$_{+1.6}$ | 16.4×$_{1.5×}$ |

**LRD is complementary to KV-cache-based accelerators.** Table 5 evaluates integrating LRD with Fast-dLLM (Wu et al., 2025), which combines KV cache and parallel decoding. Compared to the baseline decoder, Fast-dLLM alone yields substantial speedups (up to $3.0×$ on HumanEval and $11.3×$ on GSM8K) but slightly reduces accuracy. When we apply LRD on top of Fast-dLLM, accuracy not only recovers but consistently exceeds the baseline by $+1.3$ to $+2.3$ points, while the speedups *stack*, reaching $5.7×$ to $6.7×$ on HumanEval and $11.8×$ to $16.4×$ on GSM8K (an additional $1.1×$ to $2.7×$ over the corresponding Fast-dLLM variants). These results indicate that LRD is orthogonal to cache- and parallelisation-based methods and can be combined with state-of-the-art dLLM accelerators to obtain both higher accuracy and stronger overall speedups.

Table 6: Performance of Dream-Ins-7B and LLaDA-Ins-8B on Race and CNN/DailyMail. Baseline results are shown in grey, and improvements of our method are highlighted in green.

| Model | Len | Method | Race | | CNN/DailyMail | | | |
|---|---|---|---|---|---|---|---|---|
| | | | Acc | Speed | BERTScore | ROUGE-1 | ROUGE-L | Speed |
| **Dream-Ins-7B** | 256 | baseline | 81.1 | 1.0× | 85.6 | 21.7 | 15.0 | 1.0× |
| | | Ours | 82.2$_{+1.1}$ | 1.3× | 85.7$_{+0.1}$ | 22.3$_{+0.6}$ | 15.5$_{+0.5}$ | 1.3× |
| | 512 | baseline | 80.9 | 1.0× | 85.5 | 22.6 | 15.6 | 1.0× |
| | | Ours | 82.4$_{+1.5}$ | 1.4× | 85.6$_{+0.1}$ | 23.0$_{+0.4}$ | 15.9$_{+0.3}$ | 1.4× |
| | 1024 | baseline | 81.2 | 1.0× | 85.7 | 22.8 | 15.7 | 1.0× |
| | | Ours | 83.3$_{+2.0}$ | 1.7× | 85.8$_{+0.1}$ | 23.3$_{+0.5}$ | 16.5$_{+0.8}$ | 1.7× |
| **LLaDA-Ins-8B** | 256 | baseline | 67.0 | 1.0× | 82.3 | 19.9 | 13.9 | 1.0× |
| | | Ours | 68.5$_{+1.5}$ | 1.5× | 82.3 | 20.0$_{+0.1}$ | 13.9 | 1.7× |
| | 512 | baseline | 67.9 | 1.0× | 82.2 | 19.8 | 13.7 | 1.0× |
| | | Ours | 69.5$_{+1.6}$ | 1.6× | 82.4$_{+0.2}$ | 20.1$_{+0.3}$ | 13.9$_{+0.2}$ | 1.8× |
| | 1024 | baseline | 68.6 | 1.0× | 82.1 | 19.7 | 13.8 | 1.0× |
| | | Ours | 70.6$_{+2.0}$ | 1.7× | 82.3$_{+0.2}$ | 19.9$_{+0.2}$ | 14.1$_{+0.3}$ | 2.1× |

**LRD maintains general QA and summarization quality.** To verify that LRD is not over-specialised to reasoning benchmarks, we evaluate LRD on the reading comprehension dataset RACE Lai et al. (2017) and the summarization dataset CNN/DailyMail See et al. (2017) (Table 6). Across all context lengths, LRD matches or slightly improves the baseline on RACE accuracy (up to $+2.0$ points) and CNN/DailyMail metrics (BERTScore and ROUGE-1/L), while providing $1.3×$ to $2.1×$ speedups.

These results indicate that our refinement scheme preserves, and in some cases enhances, general language understanding and generation ability, rather than trading off language quality for efficiency.

## 5 Related Work

**Diffusion LLMs (dLLMs).** Diffusion models, as generative models, initially achieved significant success in continuous data domains such as image (Song et al., 2020; Ho et al., 2020; Nichol et al., 2021; Rombach et al., 2022) and speech generation (Huang et al., 2023; Yang et al., 2023). Their application in the language domain has been limited due to the discrete nature of text. One promising approach is the use of Masked Diffusion Models (MDMs) (Austin et al., 2021a; Ou et al., 2024; Shi et al., 2024; Lou et al., 2023), which represent a particular type of discrete diffusion that works with sequences through the iterative prediction of masked tokens using contextual information. Current research has concentrated on substantially expanding these MDMs. DiffuLLaMA (Gong et al., 2025), developed through continual pre-training based on LLaMA parameters, has produced diffusion Large Language Models (dLLMs) and demonstrated that dLLMs can achieve performance comparable to autoregressive models. Subsequently, higher-performance commercial dLLMs such as Mercury (Labs et al., 2025) and Gemini Diffusion (Deepmind, 2025) have been announced, along with the introduction of high-quality open-source models such as LLaDA (Nie et al., 2025; Zhu et al., 2025) and Dream (Ye et al., 2025). However, the limitations of dLLMs cannot be overlooked. Due to the lack of components analogous to KV cache and the requirement to compute results for all positions in each step, the deployment of dLLMs has consistently been constrained by inference efficiency. While reducing the number of inference steps can improve inference efficiency, this severely compromises model performance. Whether it is possible to enhance dLLMs' performance while accelerating inference remains a critical research topic for dLLMs at the current stage.

**Efficient dLLMs.** To improve dLLM inference speed while maintaining generation quality, recent works have proposed efficient dLLMs in two main directions: integrating KV cache and optimising computational load. For KV cache integration, dLLM-Cache (Liu et al., 2025) proposes a training-free adaptive caching framework addressing dual computational redundancy, specifically quasi-static prompt and dynamic response redundancy, while integrating long-interval prompt caching and V-verify mechanisms. Fast-dLLM (Wu et al., 2025) designs block-wise KV cache reuse mechanisms exploiting activation similarity in bidirectional attention, combined with confidence-aware dynamic parallel decoding. Sparse-dLLM (Song et al., 2025) combines dynamic cache eviction with sparse attention, leveraging temporal consistency of token saliency for plug-and-play inference acceleration. For computational optimisation, Prophet (Li et al., 2025b) exploits the finding that 99% of samples converge early, proposing confidence-gap-based early commitment decoding to effectively reduce decoding steps. DAEDAL (Li et al., 2025a) implements two-stage dynamic length expansion through EOS confidence prediction and low-confidence region identification, thereby enabling adaptive generation length allocation. However, all of the current works (Ben-Hamu et al., 2025; Yu et al., 2025; Ma et al., 2025; Israel et al., 2025) primarily prioritize efficiency over generation quality, largely ignoring that existing dLLMs cannot significantly outperform AR models in overall generation quality. Inspired by mixed token improvements in AR models (Zhang et al., 2025; Wang et al., 2024; Hao et al., 2024), our work emphasizes enhancing dLLMs' performance while simultaneously leveraging computed KL divergence for reliable early stopping to improve efficiency.

## 6 Conclusion

We introduced Latent Refinement Decoding, a two-stage decoding framework for diffusion language models that addresses information loss from hard masking and suboptimal commitment dynamics. By first refining global beliefs in the continuous embedding space and then entering a predictive feedback loop that progressively finalizes confident tokens under KL-based convergence monitoring, LRD preserves more information during generation and enables principled early stopping. Extensive experiments on both code generation and mathematical reasoning, QA, and summarization benchmarks demonstrate that LRD achieves consistent and significant gains in output quality and inference efficiency over standard diffusion decoding baselines Moreover, LRD is orthogonal to systems-level accelerators such as KV caching and parallel decoding: when combined with Fast-dLLM, it further improves accuracy and yields up to $2.4\times$ additional speedup over Fast-dLLM alone, highlighting its value as a flexible drop-in module for future diffusion-based LMs.

REPRODUCIBILITY STATEMENT

To ensure the reproducibility of our work, we provide complete source code as supplementary materials, including implementations for all five models (LLaDA-base, LLaDA-instruct, LLaDA-1.5, Dream-base, and Dream-instruct) evaluated on four datasets (MBPP, GSM8K, HumanEval, and MATH500), accompanied by detailed execution instructions. The model architectures are comprehensively described in Section 3, while hyperparameters for models are specified in Appendix A.1.

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

## THE USE OF LLMS

In the preparation of this manuscript, we used Large Language Models (LLMs) in a limited capacity for two specific purposes: preliminary literature survey to help identify relevant research directions and keywords during the early stages of our work, and limited language polishing to improve the clarity and grammatical correctness of certain sections in the paper. All core research ideas, theoretical contributions, experimental design, implementation, and analysis were independently conceived and conducted by the authors without LLM assistance. The LLM-generated suggestions were carefully reviewed, verified, and substantially modified by the authors before incorporation. We take full responsibility for all content presented in this paper, including any text that may have been refined with LLM assistance.

## A  EXPERIMENT DETAILS

### A.1  EXPERIMENT SETTINGS

For Base models, we follow standard few-shot settings for each benchmark: zero-shot for HumanEval, 3-shot for MBPP, 4-shot for MATH500, and 8-shot for GSM8K. For all benchmarks, we report accuracy for mathematical reasoning and pass@1 for code generation. We set the nucleus threshold to top-$p = 0.9$. The hyperparameter $r_f$ is varied between $0.1$ and $0.2$. The thresholds for stopping latent refinement and early decoding are $\tau_{\text{refine}} = 0.1$ and $\tau_{\text{decode}} = 0.1$, respectively. We cap the latent refinement stage at a maximum of $T_{\text{refine}} = 20$ steps. For LLaDA-Instruct and LLaDA-1.5 models, generation is conducted under the official semi-AR framework (Nie et al., 2025), where the sequence is divided into blocks and decoded autoregressively at the block level. Within each block, instead of the standard hard masking used in the original work, we integrate our Latent Refinement and Predictive Feedback Loop, enabling refinement of token distributions before discrete commitment. Detailed integration steps are provided in Appendix C.

**Recovering hard-assignment baselines.** The original hard-masking decoders in Dream and LLaDA can be recovered as a special case of our framework by setting the mixing ratio to zero ($r_f = 0$, so undecided positions use the plain $[\texttt{MASK}]$ embedding), disabling latent refinement ($T_{\text{refine}} = 0$), and turning off KL-based early stopping (e.g., $\tau_{\text{decode}} = 0$). Under ($r_f = 0$, $T_{\text{refine}} = 0$, $\tau_{\text{decode}} = 0$), our sampler is equivalent to the original hard-assignment baselines.

## A.2 ADDITIONAL RESULTS ON LLADA2.0-MINI-PREVIEW (16B)

Table 7: Performance of LLaDA2.0-mini-preview (16B) across benchmarks. Speed denotes relative runtime (baseline = 1.0×), where larger values indicate faster inference. Baseline results are shown in grey, and our improvements in green.

| Model | Len | Method | HumanEval | | MBPP | | GSM8K | | MATH500 | |
|---|---|---|---|---|---|---|---|---|---|---|
| | | | Acc | Speed | Acc | Speed | Acc | Speed | Acc | Speed |
| **LLaDA2.0-mini-preview** | 256 | baseline | 5.5 | 1.0× | 19.2 | 1.0× | 63.8 | 1.0× | 16.2 | 1.0× |
| | | Ours | $7.3_{+1.8}$ | 1.0× | $22.2_{+3.0}$ | 1.1× | $64.6_{+0.8}$ | 1.2× | $17.0_{+0.8}$ | 1.0× |
| | 512 | baseline | 54.3 | 1.0× | 54.7 | 1.0× | 86.5 | 1.0× | 44.6 | 1.0× |
| | | Ours | $54.9_{+0.6}$ | 1.1× | $58.2_{+3.5}$ | 1.6× | $87.6_{+1.1}$ | 2.1× | $45.8_{+1.2}$ | 1.1× |
| | 1024 | baseline | 74.2 | 1.0× | 63.2 | 1.0× | 87.7 | 1.0× | 61.2 | 1.0× |
| | | Ours | $78.7_{+4.5}$ | 1.1× | $65.7_{+2.5}$ | 2.2× | $88.9_{+1.2}$ | 3.8× | $63.2_{+2.0}$ | 1.8× |

Table 7 reports results on the larger `LLaDA2.0-mini-preview` (16B) (Nie et al., 2025) across the four benchmarks. Across all context lengths, LRD consistently improves accuracy and provides additional speedups over the baseline decoder. At length 256, LLaDA2.0 tends to produce very long outputs, which makes performance less stable in this short-context regime, but LRD still yields gains (e.g., +1.8 on HumanEval and +3.0 on MBPP); at 512 and 1024, where the model is better aligned with the available budget, LRD brings improvements of up to +4.5 points and 2.2–3.8× faster decoding, indicating that the refinement scheme scales smoothly to larger dLLMs.

## A.3 CHOICE OF CONVERGENCE METRIC

We compare several distance metrics for monitoring convergence of the latent belief state on LLaDA-1.5 with HumanEval, as shown in Figure 6. Specifically, we track step-wise distances between consecutive predictive distributions using KL divergence, Wasserstein distance, Chi-Square distance, Euclidean distance, and cosine similarity, with a 20-step latent refinement phase followed by decoding (the red vertical line marks the transition point).

KL divergence exhibits a clear and stable decreasing trend throughout both the refinement and decoding phases, providing a smooth signal that correlates well with the model's convergence behaviour. In contrast, the alternative metrics show much higher variance and noise. Based on this comparison, we adopt KL divergence as our primary convergence indicator for triggering phase transitions and early stopping in LRD.

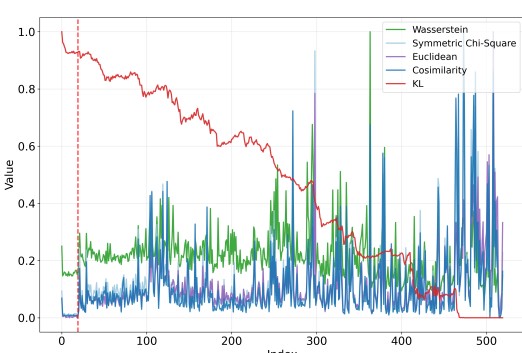

Figure 6: Step-wise predictive distributions of different distance metrics on LLaDA-1.5 (Humaneval). The red vertical line indicates where decoding begins after a fixed 20-step latent refinement.

## A.4 DECODING MULTIPLE TOKENS PER STEP

Figure 7 studies the effect of committing multiple tokens at once in Phase 2 on Dream-Base-7B with length 256. Instead of finalising only the single lowest-entropy position per step, we increase $k$ and commit the top-$k$ lowest-entropy positions simultaneously ($k \in \{1, \ldots, 5\}$). For each benchmark

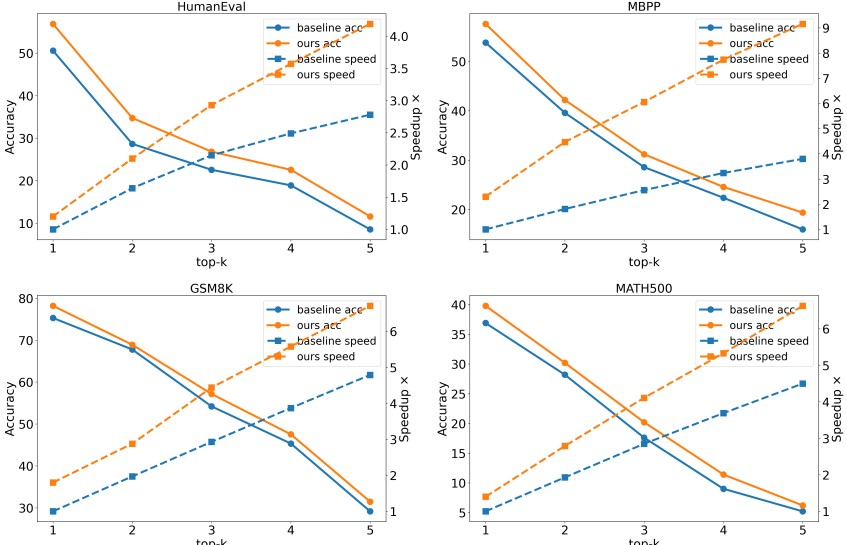

Figure 7: Comparison of Performance and Speed between our method and the Baseline when decoding multiple tokens at once on Dream-Base-7B (256 Tokens).

(HumanEval, MBPP, GSM8K, MATH500), the figure reports both accuracy and relative speed for the baseline decoder and our LRD-based decoder.

We observe a consistent pattern across all datasets: as $k$ increases, *speed* improves for both methods, but *accuracy* drops sharply, since committing many positions in parallel amplifies error propagation. Our method dominates the baseline for every $k$ in both accuracy and speed, but the overall quality–efficiency trade-off still becomes worse when $k$ is large. This suggests that naïvely increasing the number of positions decoded in parallel is not an effective way to add parallelism; instead, structural accelerators (e.g., KV-cache–based and block-wise parallel schemes) and refinement strategies like LRD provide a more principled path to achieving fast yet accurate diffusion decoding.

## A.5  COST ACCOUNTING AND QUALITY–TIME PARETO ANALYSIS

To better characterise the efficiency of LRD beyond relative "Speed" ratios, we perform a detailed cost accounting on Dream-Instruct-7B using absolute wall-clock measurements and per-step FLOP estimates (Table 8). Per step, LRD adds only negligible additional FLOPs compared to the baseline diffusion decoder, since the backbone forward pass is identical (14.14G FLOPs, 68.6 ms). However, the extra KL computation over the 152K-token vocabulary and the mixing operations introduce a memory-bound overhead, leading to a modest per-step latency increase of about 15% (from 81.8 ms to 94.3 ms). Crucially, LRD substantially reduces the number of required generation steps: the average number of steps drops from 512 to 153 (approximately 70% fewer steps), which translates into a net end-to-end speedup of roughly 2.9× in total wall-clock time (41.8s → 14.4s).

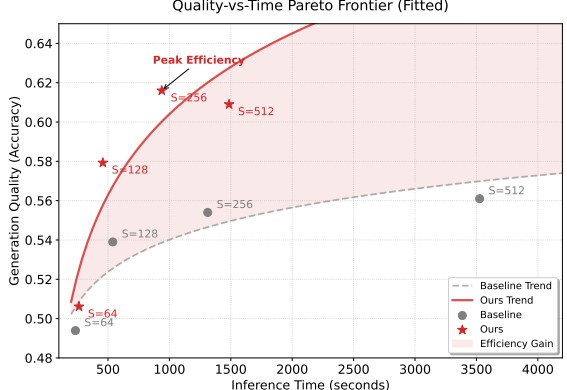

Figure 8: Quality–vs–Time Pareto analysis comparing our method with the baseline on HumanEval using Dream-Instruct under different diffusion-step budgets (2×A100 GPUs).

Table 8: **Efficiency Analysis Breakdown.** We compare the per-step theoretical cost and wall-clock time between the Baseline and our method (Dream-Instruct-7B). Although our method introduces a minor overhead per step (+15%), the significant reduction in generation steps results in a substantial total speedup.

| Per-Step Cost | FLOPs (G) | Latency | Remark / $\Delta$ |
|---|---|---|---|
| **Baseline (Standard)** | **14.14** | **81.8 ms** | **–** |
| – Backbone | 14.14 | 68.6 ms | MatMul dominated |
| – System Overhead | – | 13.2 ms | Data dispatch |
| **LRD (Proposed)** | $\sim$**14.14** | **94.3 ms** | **+15% Latency** |
| – Backbone | 14.14 | 68.6 ms | Identical to baseline |
| – KL / Mixing / Aux | $\approx 0$ | 12.6 ms | Memory-bound overhead |
| – System Overhead | – | 13.1 ms | Similar to baseline |
| Avg. Steps Needed | – | $512 \rightarrow 153$ | $\sim$**70% Reduction** |
| **Total Wall-clock** | – | **41.8s $\rightarrow$ 14.4s** | $\sim$**2.9$\times$ Speedup** |

Beyond scalar speed, we also examine the *quality–time* trade-off by varying the maximum number of diffusion steps $S \in \{64, 128, 256, 512\}$ and plotting accuracy against absolute wall-clock time on HumanEval with Dream-Instruct-7B (Figure 8). Each point in the figure corresponds to running either the baseline decoder or LRD with a given step budget $S$, and we fit smooth trend lines to visualise the resulting Pareto frontiers. Across all budgets, LRD lies on a strictly better frontier: for any given time budget it attains higher accuracy, and for any fixed accuracy level it reaches that quality faster. This shows that the small per-step overhead from KL-based refinement is more than offset by the reduced number of steps, leading to a better overall quality–efficiency trade-off.

### A.6    SCOPE OF EVALUATED DIFFUSION LM FAMILIES

Here we provide additional discussion on the scope of diffusion LM families considered in our experiments and how this relates to prior work.

**On score-based (continuous) diffusion methods.**    Score-based diffusion methods for language typically operate by embedding discrete tokens into a continuous space before applying standard continuous diffusion. While this is an interesting research direction, our work specifically focuses on discrete diffusion approaches, which models text generation directly in the discrete token space. This approach has recently demonstrated state-of-the-art performance in scalable language modeling (e.g., LLaDA, Dream).

**On other discrete diffusion frameworks.**    We acknowledge the foundational contributions of earlier discrete diffusion frameworks such as D3PM, SEED, MDLM, and BD3-LMs (Austin et al., 2021a; Lou et al., 2024; Sahoo et al., 2024; Arriola et al., 2025), as well as task-specific discrete diffusion models for summarization (Dat et al., 2025). However, our empirical evaluation focuses on *large-scale general-purpose diffusion LMs* (LLaDA, Dream) for two main reasons:

- **Scalability and generality.** Earlier discrete models (e.g., the original D3PM and SEED baselines, MDLM, BD3-LMs) are typically smaller than 350M parameters and are often designed for specific tasks or benchmarks, frequently requiring fine-tuning or task-specific adaptation. In contrast, LLaDA (7B) and Dream (8B) represent a new generation of diffusion LMs with strong zero-shot general, coding, and reasoning capabilities, and have quickly become de facto testbeds for subsequent dLLM acceleration and control methods (Wu et al., 2025; Liu et al., 2025; Wang et al., 2025).

- **Training-free objective.** Our proposed method, LRD, is a *training-free decoding strategy* designed to unlock the potential of these general-purpose models without task-specific retraining. Evaluating LRD primarily on small, specialised diffusion models would be less representative of its intended use case, namely improving the quality–efficiency trade-off in general-purpose inference for large diffusion LMs.

## B  DERIVATION OF THE TRUE POSTERIOR IN THE MASKING PROCESS

We derive Eq. 5 for the true posterior distribution in the absorbing masking forward process. For each position $i$, the forward process is defined as

$$\Pr(x_t^{(i)} = x_0^{(i)} \mid x_0^{(i)}) = \alpha_t^*, \qquad \Pr(x_t^{(i)} = \texttt{[MASK]} \mid x_0^{(i)}) = 1 - \alpha_t^*,$$

with $(\alpha_t^*)_{t=0}^T$ monotonically decreasing. Thus each token can only either remain as its original value $x_0^{(i)}$ or transition to the special token $\texttt{[MASK]}$. By Bayes' rule,

$$q^*(x_{t-1}^{(i)} \mid x_t^{(i)} = \texttt{[MASK]}, x_0^{(i)}) = \frac{\Pr(x_t^{(i)} = \texttt{[MASK]} \mid x_{t-1}^{(i)}, x_0^{(i)}) \Pr(x_{t-1}^{(i)} \mid x_0^{(i)})}{\Pr(x_t^{(i)} = \texttt{[MASK]} \mid x_0^{(i)})}. \qquad (8)$$

There are two possible values for $x_{t-1}^{(i)}$:

- The probability of $x_{t-1}^{(i)} = x_0^{(i)}$ is $\alpha_{t-1}^*$, and transitioning to mask at step $t$ occurs with probability $1 - \frac{\alpha_t^*}{\alpha_{t-1}^*}$. Hence the joint probability is $\alpha_{t-1}^* - \alpha_t^*$.

- The probability of $x_{t-1}^{(i)} = \texttt{[MASK]}$ is $1 - \alpha_{t-1}^*$, and once masked, the token remains masked with probability 1. Hence the joint probability is $1 - \alpha_{t-1}^*$.

The marginal probability of being masked at step $t$ is $Pr(x_t^{(i)} = \texttt{[MASK]} \mid x_0^{(i)}) = 1 - \alpha_t^*$. So we obtain

$$q^*(x_{t-1}^{(i)} \mid x_t^{(i)} = \texttt{[MASK]}, x_0^{(i)}) = \frac{\alpha_{t-1}^* - \alpha_t^*}{1 - \alpha_t^*} \delta_{x_0^{(i)}} + \frac{1 - \alpha_{t-1}^*}{1 - \alpha_t^*} \delta_{\texttt{[MASK]}}.$$

## C  INTEGRATION WITH SEMI-AR FRAMEWORK

In the semi-AR setting in LLaDA (Nie et al., 2025), a sequence of length $L$ is partitioned into $B$ blocks $\{b_1, b_2, ..., b_B\}$. While their original work uses standard hard masking within each block, we apply soft embeddings as follows:

For each block $b_i$ conditioned on previously generated blocks $\{b_1, ..., b_{i-1}\}$:

1. **Soft Refinement:** Initialise positions in $b_i$ with $\texttt{[MASK]}$ embeddings, then apply soft embedding refinement (Equation 3) until convergence.

2. **Progressive Decoding:** Use the converged soft embeddings to guide token selection within the block.

## D  STABILITY ANALYSIS OF MIXED EMBEDDING UPDATES

Our method operates in the embedding space rather than the discrete token space. At each timestep $t$, we maintain a set of *soft embeddings* $\mathcal{E}_t = \{\tilde{\mathbf{e}}_t^{(1)}, \dots, \tilde{\mathbf{e}}_t^{(L)}\}$ defined as

$$\tilde{\mathbf{e}}_t^{(i)} = (1 - \alpha_t^{(i)}) \cdot \mathbf{e}_{\texttt{[MASK]}} + \alpha_t^{(i)} \cdot \sum_{v \in \mathcal{T}_t^{(i)}} \bar{p}_{t+1}^{(i)}(v) \cdot \mathbf{e}_v, \qquad (9)$$

where $\mathbf{e}_{\texttt{[MASK]}}$ denotes the $\texttt{[MASK]}$ embedding, $\mathbf{e}_v$ denotes the embedding of token $v$, $\mathcal{T}_t^{(i)}$ is the top-$p$ nucleus set at position $i$, and $\bar{p}_{t+1}^{(i)}(v)$ is the renormalised predicted distribution over the nucleus set at position $i$.

To analyse stability, an ideal approach would be to examine the Jacobian of the update operator through its spectral radius. However, in practice this is intractable: transformer structures involve many linear and nonlinear components (layer normalisation, residual connections, multi-head attention), making it nearly impossible to provide a formal global analysis. The effective Jacobian inherits the complexity of the underlying transformer, and its spectral radius (or even its spectral norm) may be large and

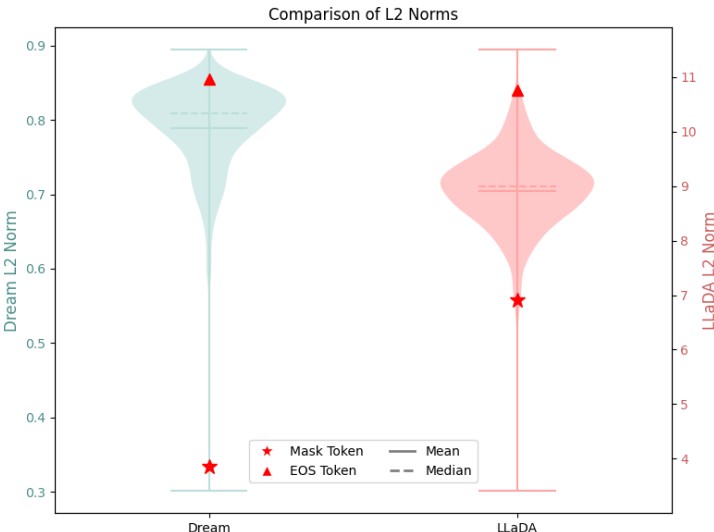

Figure 9: Distribution of L2 norms for token embeddings in Dream and LLaDA models.

not easily bounded. As a result, although the iteration often stabilises empirically, a rigorous global convergence guarantee cannot be obtained.

Therefore, in this section, we follow the discussion from existing work (Yudin et al., 2025; Hu et al., 2024; Dasoulas et al., 2021) and focus on *local* Lipschitz continuity. This analysis considers only a single self-attention layer without any other operators and provides intuition to support our method and explain empirical results.

Specifically, the local Lipschitz bound suggests that for all soft embedding $e_t$ within an $\epsilon$-ball at original point (i.e. $\|e_t\| \leq \epsilon$), where $\epsilon$ in fact bounds the maximum norm of embeddings, the following inequality holds after one-layer self-attention mapping:

$$\|\mathbf{e}_{t+1}^s - \mathbf{e}_t^s\|_2 \leq K\|\mathbf{e}_{t+1} - \mathbf{e}_t\|_2, \tag{10}$$

where $\mathbf{e}_t^s$ is the output of $\mathbf{e}_t$ after one-layer self-attention mapping, $K$ is the local Lipschitz constant. Following Hu et al. (2024), we approximate $K$ in the form

$$K(\epsilon) \ \propto \ c\, \|\mathbf{W}_h^V\|_2\, \|\mathbf{W}_h^Q(\mathbf{W}_h^K)^\top\|_2\, \epsilon^2, \tag{11}$$

depends on the local norm $\epsilon$, with query, key, and value matrices $\mathbf{W}_h^Q, \mathbf{W}_h^K, \mathbf{W}_h^V$, and a scaling constant $c$.

The ideal outcome of such a mapping would be a contraction, i.e. $K \leq 1$, which ensures that differences shrink across layers. However, in transformer blocks the large parameter norms often make this condition difficult to satisfy. Since $\mathbf{W}_h^Q, \mathbf{W}_h^K$, and $\mathbf{W}_h^V$ are fixed for a pretrained model, stability in practice relies on keeping $\epsilon$ sufficiently small, which is under our control. This motivates us to restrict the update within a small $\epsilon$-ball neighbourhood of the [MASK] embedding, which can be taken as a reference point near the origin. For comparison, in *Dream* (Ye et al., 2025), while the [MASK] embedding has a very small $\ell_2$ norm of 0.3340 in 3,584 dimensions (corresponding to a per-dimension RMS of about 0.0055), regular token embeddings are much larger. Figure 9 shows the distribution of L2 norms across all token embeddings in both Dream and LLaDA models, confirming that the [MASK] token consistently exhibits substantially lower norms than regular tokens, validating our design choice to use [MASK] as a stable low-norm reference point for mixed embedding updates.

To connect this bound back to the embedding updates, we require $\tilde{\mathbf{e}}_t^{(i)}$ and $\tilde{\mathbf{e}}_{t+1}^{(i)}$ to lie within an $\epsilon$-ball at origin, which requires a very small $\epsilon$. Since both are formed as weighted sums of the [MASK] embedding and candidate token embeddings (Equation. 9), a straightforward way to reduce this distance is to bound the mixing coefficient $\alpha_t^{(i)}$. Intuitively, this means the search for efficient

mixed embeddings remains close to the `[MASK]` token, with exploration constrained to a small neighbourhood. In this way, the iterative updates remain within a contraction-like region, which empirically yields stable predictive distributions.

To simplify, we introduce a base rate $r_f$ and set $\alpha_t^{(i)} = r_f \cdot \hat{H}_{t+1}^{(i)}$, where $\hat{H}_{t+1}^{(i)} \in [0, 1]$ is the normalised entropy. Since $\max_i \alpha_t^{(i)} \leq r_f$, ensuring the difference is within $\epsilon$ reduces to choosing a sufficiently small $r_f$. Empirically, we find that the method is stable and effective when $r_f$ is small, but fails to converge for large $r_f$ (see Figure 4).

We further evaluate the stability of output embeddings before the logit prediction step across adjacent timesteps. Since the token space is sparse and high-dimensional, we use the KL divergence as the metric. This reveals clear convergence during the latent refinement phase when $r_f$ is small, even after deep iteration with multi-layer self-attention in a transformer (see Figure 3).

Another observation that implicitly supports our claim is the case of top-$p$ selection. If $p$ is set very small, only a few candidate tokens contribute to the weighted sum $\sum_{v \in \mathcal{T}_t^{(i)}} \bar{p}_{t+1}^{(i)}(v) \mathbf{e}_v$. Even without an explicit scaling factor such as $\alpha_t^{(i)}$, restricting the support of the soft embedding effectively yields a small $\epsilon$, which can help stabilise the updates. This explains why our method maintains reasonable performance even under extreme top-$p$ settings (see Figure 5).

In summary, although a rigorous global convergence guarantee for mixed embedding iterations is intractable due to the nonlinear, high-capacity nature of transformers, our local Lipschitz analysis provides useful theoretical insight. Together with empirical validation, this suggests that while strict guarantees remain challenging, the proposed method is practically stable and effective for reasoning with diffusion LLMs.

