# OpenReview forum: "Latent Refinement Decoding: Enhancing Diffusion-Based Language Models by Refining Belief States"
_ICLR.cc/2026/Conference — ICLR 2026 Conference Withdrawn Submission_

### Official Review · Reviewer_b4sQ · 2025-10-30

**Soundness:** 3
**Presentation:** 3
**Contribution:** 3
**Rating:** 6
**Confidence:** 3

**Summary:**

This paper proposes Latent Refinement Decoding (LRD), a two-stage decoding framework for diffusion-based language models that addresses information loss from hard masking and inefficient convergence. The method operates by: (1) Phase 1 - maintaining masked positions as distributional mixtures of predicted tokens and mask embeddings in continuous space (Latent Refinement), and (2) Phase 2 - progressively finalizing confident tokens while retaining uncertain ones as soft embeddings (Predictive Feedback Loop). KL-divergence is used to monitor convergence and enable adaptive early stopping. Experiments on LLaDA and Dream models across coding (HumanEval, MBPP) and reasoning (GSM8K, Math500) benchmarks demonstrate accuracy improvements of 0.7-6.3 points with speedups up to 10.6×.

**Strengths:**

1. Identifies specific failure modes of hard masking (information loss, infinite KL divergence under misprediction) with mathematical justification
2. KL-divergence monitoring provides an adaptive, task-dependent stopping criterion rather than fixed iteration counts
3. 5 model variants × 4 benchmarks × 3 sequence lengths = 60 experimental configurations with consistent improvements
4. Tables 2-4 and Figures 4-5 systematically validate each component's contribution

**Weaknesses:**

1. Only evaluated on two diffusion LM families (LLaDA, Dream). Generalization to other discrete diffusion approaches (e.g., score-based methods, D3PM) is unclear.
2. While HumanEval shows strong gains, improvements on GSM8K and MATH500 are sometimes quite small (e.g., +0.7 points).
3. Related work mentions dLLM-Cache, Fast-dLLM, Prophet, etc., but no direct comparisons are provided.

**Questions:**

1. How should practitioners set hyperparameters for new tasks without grid search?
2. How does LRD compare directly with dLLM-Cache, Fast-dLLM, or Prophet? Can they be combined?
3. Table 4 shows removing latent refinement sometimes helps, when is Phase 1 actually beneficial?
4. How does the method scale beyond 1024 tokens?

---

> ### Author Response · Authors · 2025-11-23
> **Response to Reviewer b4sQ (Part 1/3)**
>
> We are grateful to Reviewer b4sQ for the constructive feedback. Please find our detailed responses below.
>
> ---
>
>
> **1.Evaluate other diffusion model families**
> >Only evaluated on two diffusion LM families (LLaDA, Dream). Generalization to other discrete diffusion approaches (e.g., score-based methods, D3PM) is unclear.
>
> Thanks for your comments. We clarify our scope and positioning below.
>
> **On score-based methods:** Score-based diffusion methods[1,2] for language typically operate by embedding discrete tokens into a continuous space before applying standard continuous diffusion. While this is an interesting research direction, our work specifically focuses on discrete diffusion approaches, which models text generation directly in the discrete token space. This approach has recently demonstrated state-of-the-art performance in scalable language modeling (e.g., LLaDA, Dream).
>
> **On other discrete diffusion frameworks:** We acknowledge the foundational contributions of earlier discrete frameworks such as D3PM[3], SEED[4], MDLM[5], BD3-LMs[6], as well as task-specific discrete diffusion models for summarization[7]. However, our evaluation focuses on **Large-Scale General-Purpose Diffusion LMs** for the following reasons:
>
> - **Scalability and Generality**: As shown in Table, earlier discrete models (like the original D3PM and SEED baselines) are typically **small (<350M parameters)** and non-general purpose, often requiring fine-tuning for new tasks. In contrast, LLaDA (7B) and Dream (8B) represent a new generation of models with strong zero-shot general, coding, and reasoning capabilities. They have quickly become de facto testbeds for follow-up work [8,9,10]
> - **Training-Free Objective**: Our proposed method, LRD, is a training-free decoding strategy designed to unlock the potential of these general-purpose models without task-specific adaptation. Evaluating LRD on smaller, specialized models would not accurately reflect its utility in general-purpose inference scenarios.
>
> We have added this discussion to Appendix A.6(line 890).
>
> | Model Family  | Representative Models | size  | General LLMs? | Requires Fine-tuning for New Tasks? |
> |-|-|-|-|-|
> | Early Discrete Frameworks | D3PM[3], SEED[4], MDLM[5], BD3-LMs[6], CrossMamba[7]   | < 350M  | No (Task-Specific)| Yes|
> | Large-Scale Diffusion LMs | LLaDA, Dream (Our Focus)  | 7B–8B   | Yes     | No  |
> |
>
> [1] Li, Xiang, et al. "Diffusion-lm improves controllable text generation." Advances in neural information processing systems 35 (2022): 4328-4343.
>
> [2] Han, Xiaochuang, Sachin Kumar, and Yulia Tsvetkov. "Ssd-lm: Semi-autoregressive simplex-based diffusion language model for text generation and modular control." Proceedings of the 61st Annu al Meeting of the Association for Computational Linguistics (Volume 1: Long Papers). 2023.
>
> [3] Austin, Jacob, et al. "Structured denoising diffusion models in discrete state-spaces." Advances in neural information processing systems 34 (2021): 17981-17993.
>
> [4] Lou, Aaron, Chenlin Meng, and Stefano Ermon. "Discrete diffusion modeling by estimating the ratios of the data distribution." arXiv preprint arXiv:2310.16834 (2023).
>
> [5] Sahoo, Subham, et al. "Simple and effective masked diffusion language models." Advances in Neural Information Processing Systems 37 (2024): 130136-130184.
>
> [6] Arriola, Marianne, et al. "Block diffusion: Interpolating between autoregressive and diffusion language models." arXiv preprint arXiv:2503.09573 (2025).
>
> [7] Do, Duc Anh, Luu Anh Tuan, and Wray Buntine. "Discrete diffusion language model for efficient text summarization." Findings of the Association for Computational Linguistics: NAACL 2025. 2025.
>
> [8] Wu, Chengyue, et al. "Fast-dllm: Training-free acceleration of diffusion llm by enabling kv cache and parallel decoding." arXiv preprint arXiv:2505.22618 (2025).
>
> [9] Liu, Zhiyuan, et al. "dllm-cache: Accelerating diffusion large language models with adaptive caching." arXiv preprint arXiv:2506.06295 (2025).
>
> [10] Wang, Xu, et al. "Diffusion llms can do faster-than-ar inference via discrete diffusion forcing." arXiv preprint arXiv:2508.09192 (2025).

---

> ### Author Response · Authors · 2025-11-23
> **Response to Reviewer b4sQ (Part 2/3)**
>
> **2.Consistency of improvements across benchmarks**
> >While HumanEval shows strong gains, improvements on GSM8K and MATH500 are sometimes quite small (e.g., +0.7 points).
>
> We agree that the absolute gains on GSM8K and MATH500 are smaller than on HumanEval, and this is largely because our dLLM baselines are already very strong on these reasoning benchmarks, leaving limited headroom. In this high-accuracy regime, our improvements ranging from +0.7 to +3.8 points represent meaningful reductions in error rate. For example, on GSM8K, improving from 82.3% to 83.7% (+1.4 points) reduces the error rate from 17.7% to 16.3%: a **7.9% relative error reduction**.
>
> Crucially, these improvements are **stable and monotonic**: across all model families (Dream, LLaDA), model variants (Base, Instruct), and sequence lengths (256, 512, 1024), LRD never degrades performance on reasoning tasks. Moreover, these quality improvements come with substantial speedups (1.4-5.5×).
>
>
> We have also evaluated LRD on general language tasks (RACE for QA, CNN/DailyMail for summarisation, Table 6), showing consistent improvements. Taken together, LRD moves dLLMs to a strictly better quality-efficiency operating point across diverse tasks.
>
>
>
>
> **3.Practical guidance for hyperparameter selection**
> >How should practitioners set hyperparameters for new tasks without grid search?
>
> LRD is designed to work with minimal tuning, so we do not rely on large grid searches in our experiments. In practice, we keep most hyperparameters (top-p, KL thresholds $\(\tau_{\text{refine}}, \tau_{\text{decode}}\$) fixed across models and tasks. The sensitivity analyses in Figures 4 and 5 show that performance is quite stable under moderate changes of the mixing and selection parameters. For new tasks, our recommendation is therefore to start from the default hyperparameters reported in the paper and, only if the task is substantially different (e.g., very long contexts or a markedly new domain), perform a light sanity check on a small validation split rather than an exhaustive grid search.
>
>
> **4.Comparison and compatibility with existing acceleration methods**
> >How does LRD compare directly with dLLM-Cache, Fast-dLLM, or Prophet? Can they be combined?
>
> dLLM-Cache, Fast-dLLM, and Prophet target complementary aspects of dLLM efficiency: dLLM-Cache and Fast-dLLM focus on KV-cache reuse and parallel decoding, while Prophet focuses on step scheduling and early commitment. In contrast, LRD changes the denoising schedule itself, namely how information is propagated and when tokens are committed, without modifying the underlying transformer architecture or cache mechanism, which makes it naturally orthogonal to these methods.
> |Model|Method|Humaneval Acc|Humaneval Speed|GSM8K Acc|GSM8K Speed|
> |-|-|-|-|-|-|
> |**Dream-base-7B**|Baseline|54.3|1.0×|76.0|1.0×|
> ||Fast-dLLM(prefix cache)|54.2(−0.1)|2.4×|74.5(−1.5)|4.8×|
> ||Fast-dLLM(prefix cache)+Ours|56.0(+1.7)|5.7×|77.6(+1.6)|13.1×|
> ||Fast-dLLM(dual cache)|54.2(−0.1)|3.0×|74.2(−1.8)|6.0×|
> ||Fast-dLLM(dual cache)+Ours|56.3(+2.0)|6.7×|77.3(+1.3)|15.2×|
> |**LLaDA-Ins-8B**|Baseline|43.5|1.0×|77.1|1.0×|
> ||Fast-dLLM(prefix cache)|43.4(−0.1)|2.8×|76.8(−0.3)|10.4×|
> ||Fast-dLLM(prefix cache)+Ours|45.8(+2.3)|3.8×|78.8(+1.7)|11.8×|
> ||Fast-dLLM(dual cache)|43.2(−0.3)|3.2×|76.7(−0.4)|11.3×|
> ||Fast-dLLM(dual cache)+Ours|45.7(+2.2)|4.7×|78.7(+1.6)|16.4×|
> |
>
> To make this concrete, we implemented a direct comparison and combination with Fast-dLLM (which integrates KV cache and confidence-aware parallel decoding). As shown in the table above, Fast-dLLM alone delivers substantial speedups but slightly reduces accuracy (e.g., Dream GSM8K: 76.0 → 74.5/74.2, −1.5/−1.8). When we apply LRD on top of Fast-dLLM, accuracy becomes **higher than both the baseline and Fast-dLLM alone** (e.g., Dream GSM8K: 77.3–77.6, i.e., +1.3–1.6 over baseline and +2.8–3.4 over Fast-dLLM), while speedups stack, reaching up to **15.2×–16.4×**. This indicates that LRD is synergistic with state-of-the-art decoding/selection variants rather than a competing alternative.  We have added the new table in the extended ablation study (Line 457)

---

> ### Author Response · Authors · 2025-11-23
> **Response to Reviewer b4sQ (Part 3/3)**
>
> **5.When latent refinement is actually beneficial**
> >Table 4 shows removing latent refinement sometimes helps, when is Phase 1 actually beneficial?
>
> Thank you for this important clarification. The key is to distinguish between *speed* and *overall benefit*.
>
> Table 4 reports only compute metrics (Speed and $E_{token}$), so "w/o latent refinement" appears slightly faster by simply skipping Phase 1. However, this comes at two costs:
> - Lower accuracy: Table 3 shows removing latent refinement consistently reduces accuracy
> - Less efficient convergence: Higher $E_{token}$ in most cases
>
> Full LRD achieves both better accuracy *and* substantial speedups over baseline. Trading quality for marginal speed gain is not the design goal of LRD.
>
> **6.How does the method scale beyond 1024 tokens?**
>
>
> Following the reviewer's suggestion, we have further verified that LRD scales to sequences of 2048 tokens:
>
>
> |Model|Len|Method|HumanEval Acc|HumanEval Speed|MBPP Acc|MBPP Speed|GSM8K Acc|GSM8K Speed|MATH500 Acc|MATH500 Speed|
> |---|---|---|---|---|---|---|---|---|---|---|
> |Dream-Ins-7B|256|baseline|55.4|1.0×|57.4|1.0×|80.8|1.0×|37.9|1.0×|
> ||256|Ours|61.6(+6.2)|1.4×|59.4(+2.0)|2.4×|83.0(+2.2)|1.4×|40.6(+2.7)|1.1×|
> ||512|baseline|56.1|1.0×|56.7|1.0×|80.2|1.0×|38.6|1.0×|
> ||512|Ours|60.9(+4.8)|2.9×|58.8(+2.1)|4.6×|82.7(+2.5)|3.6×|41.8(+3.2)|1.2×|
> ||1024|baseline|56.0|1.0×|57.3|1.0×|81.3|1.0×|40.1|1.0×|
> ||1024|Ours|61.0(+5.0)|9.3×|59.0(+1.7)|10.6×|83.5(+2.2)|5.5×|43.9(+3.8)|1.7×|
> ||2048|baseline|56.1|1.0×|57.1|1.0×|81.1|1.0×|41.1|1.0×|
> ||2048|Ours|60.8(+4.7)|17.3×|59.1(+2.0)|23.6×|83.6(+2.5)|11.6×|44.3(+3.2)|3.6×|
> |
>
> The results confirm the scaling trend: LRD maintains consistent performance improvements while achieving even higher speedups. This validates our conclusion that for applications requiring longer contexts, LRD scales naturally as long as the underlying dLLM supports the target length.

---

### Official Review · Reviewer_Q1tv · 2025-11-01

**Soundness:** 3
**Presentation:** 3
**Contribution:** 3
**Rating:** 6
**Confidence:** 3

**Summary:**

The paper proposes Latent Refinement Decoding (LRD) for diffusion-based language models (dLLMs). LRD replaces purely hard masking/commitment with a two-phase process: (i) a latent refinement phase that mixes the [MASK] embedding with an entropy-normalized expectation of top-p token embeddings (carrying forward uncertainty rather than discarding it), followed by (ii) a predictive feedback phase that progressively commits low-entropy positions while keeping uncertain positions soft, with KL-divergence dynamics used to trigger phase transition and early stopping. On GSM8K, MATH500, MBPP, and HumanEval, LRD yields consistent accuracy improvements and notable speedups.

**Strengths:**

- Clear, intuitive motivation. The paper correctly identifies two practical issues in current discrete diffusion decoding—information loss from hard remasking and premature commitment—and directly addresses both with a principled soft-to-hard schedule. The mechanics of mixing token and mask embeddings are described crisply and tied to entropy.
- Method design is coherent. The KL-based stability monitor is a sensible criterion both for the phase switch and for early stopping; it connects the algorithm’s control flow to a measurable distributional convergence signal.
- Empirical results are broad and consistent. Gains appear across two families (Dream-7B, LLaDA-8B/1.5), multiple lengths (256–1024), and both reasoning and coding. The tables show +1–6 points accuracy and sizable speedups, especially at long contexts.
- Ablations are thoughtful. The paper isolates the contribution of hot-start refinement, mixed embeddings, and early stopping; it also explores nucleus size/top-p and max mix ratio. This helps validate that mixing drives much of the gain, while early stopping drives most speedup.

**Weaknesses:**

- Limited baselines beyond “standard” discrete diffusion decoding. While the paper thoroughly compares to its own “hard” baseline, I’d like to see head-to-head against stronger decoding/selection variants for dLLMs under identical compute budgets. Some are discussed in related work, but the empirical comparison is not fully exhaustive.
- Scope: reasoning/coding only. Results are on GSM8K, MATH500, HumanEval, MBPP. It would be helpful to see a general-purpose generation benchmark (e.g., long-form QA, summarization) to test whether the latent mixing compromises stylistic fidelity or introduces artifacts in open-ended text.
- Theoretical story is partial. The KL-monitor is motivated (and the hard-mask KL blow-up is a nice point), but the paper does not provide deeper convergence guarantees for the two-phase schedule beyond a qualitative justification. A simplified setting with a contraction argument or fixed-point analysis would strengthen the case.
- Cost accounting and fairness. Reported “Speed” is relative runtime (baseline = 1.0×), but details of per-step FLOPs and wall-clock contributions of KL computation, extra forward passes in latent mixing, and cache behaviors (if any) are not fully disentangled. A standardized tokens-per-second at equal quality or quality-at-fixed-time curve would make the speed claims crisper.

**Questions:**

- How does LRD interact with KV caching or block-reuse schemes for dLLMs? Does the latent mixing impede cache reusability, or can you cache attention over mixed embeddings safely?

- Can you show quality-vs-time Pareto curves across decoding policies to confirm LRD’s advantage at matched budgets?

- For the semi-AR block setting, do beliefs transfer between blocks (e.g., previous block’s soft state informing the next), or is each block re-initialized?

---

> ### Author Response · Authors · 2025-11-23
> **Response to Reviewer Q1tv (Part 1/2)**
>
> We sincerely appreciate Reviewer Q1tv’s insightful feedback. Please see our detailed responses below.
>
> ---
>
>
> **1. Comparison with Other Decoding Variants & Interaction with KV Caching**
>
> Thank you for raising these points. Our goal in this work is to improve the *quality–speed trade-off of the denoising schedule itself* for dLLMs, in a way that is orthogonal and complementary to existing efficiency methods (KV caching, parallel decoding, etc.). Earlier works such as dLLM-Cache, Fast-dLLM, and Sparse-dLLM mainly focus on reducing per-step or per-token compute via cache reuse and structural sparsity. In contrast, LRD changes how information is propagated and when tokens are committed, without modifying the underlying transformer or training procedure.
>
> To further demonstrate the potential of integrating our method with existing approaches, we have added experiments that **explicitly combine LRD with Fast-dLLM** (which integrates KV cache and confidence-aware parallel decoding).
> |Model|Method|Humaneval Acc|Humaneval Speed|GSM8K Acc|GSM8K Speed|
> |-|-|-|-|-|-|
> |**Dream-base-7B**|Baseline|54.3|1.0×|76.0|1.0×|
> ||Fast-dLLM(prefix cache)|54.2(−0.1)|2.4×|74.5(−1.5)|4.8×|
> ||Fast-dLLM(prefix cache)+Ours|56.0(+1.7)|5.7×|77.6(+1.6)|13.1×|
> ||Fast-dLLM(dual cache)|54.2(−0.1)|3.0×|74.2(−1.8)|6.0×|
> ||Fast-dLLM(dual cache)+Ours|56.3(+2.0)|6.7×|77.3(+1.3)|15.2×|
> |**LLaDA-Ins-8B**|Baseline|43.5|1.0×|77.1|1.0×|
> ||Fast-dLLM(prefix cache)|43.4(−0.1)|2.8×|76.8(−0.3)|10.4×|
> ||Fast-dLLM(prefix cache)+Ours|45.8(+2.3)|3.8×|78.8(+1.7)|11.8×|
> ||Fast-dLLM(dual cache)|43.2(−0.3)|3.2×|76.7(−0.4)|11.3×|
> ||Fast-dLLM(dual cache)+Ours|45.7(+2.2)|4.7×|78.7(+1.6)|16.4×|
> |
>
> Fast-dLLM alone provides substantial speedups but introduces a slight accuracy drop (–0.1 to –1.8 points), whereas incorporating LRD on top of Fast-dLLM not only recovers but improves accuracy by +1.3 to +2.3 points over the baseline, with the speed benefits compounding to reach as high as 16.4×.
> These results show that LRD is synergistic with state-of-the-art decoding/selection variants rather than being an alternative that must be used in isolation.
>
> Regarding KV caching and block reuse, LRD does **not** interfere with cache reusability: latent mixing is applied at the *input embedding level* before the transformer layers, whereas KV caches store intermediate key/value states inside the network. Once the mixed embeddings are fed into the transformer, attention and KV caching operate exactly as in the base dLLM or Fast-dLLM implementation. Empirically, the combined Fast-dLLM + LRD variants retain the expected KV-cache speedups while further improving convergence, confirming that attention over mixed embeddings can be cached safely. We have added the new table and an explicit discussion of this compatibility in the extended ablation study (Line 457).
>
>
>
>
> **2. Broader Task Coverage**
>
> > Scope: reasoning/coding only. It would be helpful to see a general-purpose generation benchmark (e.g., long-form QA, summarization) to test whether the latent mixing compromises stylistic fidelity or introduces artifacts in open-ended text.
>
> To address this concern, we evaluate LRD on open-ended generation tasks beyond reasoning/coding: RACE [1] for reading comprehension and CNN/DailyMail [2] for summarisation. Results are shown in the table below:
>
> | Model | Len | Method | RACE: Acc | RACE: Speed | CNN/DM: BERTScore | CNN/DM: ROUGE-1 | CNN/DM: ROUGE-L | CNN/DM: Speed |
> |-|-|-|-|-|-|-|-|-|
> | Dream-Ins-7B | 256 | baseline | 81.1 | 1.0× | 85.6 | 21.7 | 15.0 | 1.0× |
> | | | Ours | 82.2 (+1.1) | 1.3× | 85.7 (+0.1) | 22.3 (+0.6) | 15.5 (+0.5) | 1.3× |
> | | 512 | baseline | 81.9 | 1.0× | 85.5 | 22.6 | 15.6 | 1.0× |
> | | | Ours | 82.4 (+0.5) | 1.4× | 85.6 (+0.1) | 23.0 (+0.4) | 15.9 (+0.3) | 1.4× |
> | | 1024 | baseline | 81.7 | 1.0× | 85.7 | 22.8 | 15.7 | 1.0× |
> | | | Ours | 82.7 (+1.0) | 1.7× | 85.8 (+0.1) | 23.3 (+0.5) | 16.5 (+0.8) | 1.7× |
> | LLaDA-Ins-8B | 256 | baseline | 67.0 | 1.0× | 82.3 | 19.9 | 13.9 | 1.0× |
> | | | Ours | 68.5 (+1.5) | 1.5× | 82.3 | 20.0 (+0.1) | 13.9 | 1.7× |
> | | 512 | baseline | 67.9 | 1.0× | 82.2 | 19.8 | 13.7 | 1.0× |
> | | | Ours | 69.5 (+1.6) | 1.6× | 82.4 (+0.2) | 20.1 (+0.3) | 13.9 (+0.2) | 1.8× |
> | | 1024 | baseline | 68.6 | 1.0× | 82.1 | 19.7 | 13.8 | 1.0× |
> | | | Ours | 70.6 (+2.0) | 1.7× | 82.3 (+0.2) | 19.9 (+0.2) | 14.1 (+0.3) | 2.1× |
> |
>
> Results show that LRD consistently improves quality on both tasks while maintaining speedups (1.3×-2.1×). For summarisation, improvements in ROUGE scores demonstrate that latent mixing does not compromise stylistic fidelity or introduce artifacts. The consistent BERTScore improvements further confirm semantic preservation in open-ended generation.We have added the new table in the extended ablation study (Line 482).
>
>
> [1] Lai et al. RACE: Large-scale ReAding Comprehension Dataset From Examinations. EMNLP 2017.
>
> [2] See et al. Get To The Point: Summarization with Pointer-Generator Networks. ACL 2017.

---

> ### Author Response · Authors · 2025-11-23
> **Response to Reviewer Q1tv (Part 2/2)**
>
> **3. Convergence Guarantees**
>
> > A simplified setting with a contraction argument or fixed-point analysis would strengthen the case.
>
> We appreciate the reviewer’s concern. We assumed a simplified setting (considers only a single self-attention layer without any other operators), using local Lipschitz analysis to establish stability conditions for soft embedding updates. This convergence actually depends on local Lipschitz continuity.
>
> The convergence condition requires that input token norms should be bounded within a neighborhood of the origin. Therefore, we verified whether the $\texttt{[MASK]}$ token satisfies this condition: as shown in Figure 9 (Appendix D, line 990), $\texttt{[MASK]}$ embeddings have substantially lower L2 norms compared to regular tokens. By setting the mixing coefficient $\alpha_t$ sufficiently small (controlled by $r_f$), we ensure that mixed embeddings remain within this contraction-like region, guaranteeing stable convergence.
>
> While a rigorous global convergence proof remains intractable for full transformer blocks, our local Lipschitz analysis provides theoretical grounding for the empirical stability observed in practice. A complete theoretical discussion can be found in Appendix D(Line 956).
>
>
>
>
> **4. Cost Accounting and Fairness**
>
> > Reported “Speed” is relative runtime (baseline = 1.0×), but details of per-step FLOPs and wall-clock contributions of KL computation, extra forward passes in latent mixing, and cache behaviors (if any) are not fully disentangled. A standardized tokens-per-second at equal quality or quality-at-fixed-time curve would make the speed claims crisper.
>
> We have conducted comprehensive cost accounting using absolute wall-clock time and plotted Quality-vs-Time Pareto frontiers to address this concern in Table 8 (Appendix A.5 line 864)
>
> - Per-Step Cost Breakdown:
> |Component Breakdown|FLOPs(G)|Time|Note|
> |-|-|-|-|
> |**Baseline(PerStep)**|14.14G|81.8ms||
> |Backbone Inference|14.14G|68.6ms|Dominated by MatMul|
> |System Overhead|||Data movement/Dispatch|
> |**Ours(PerStep)**|~14.14G|94.3ms|~15% latency increase|
> |Backbone Inference|14.14G|68.6ms|Identical to baseline|
> |KL/Mixing/Aux|Negligible*|12.6ms|Memory-bound overhead|
> |System Overhead|||Similar to baseline|
> |**Total Efficiency**||||
> |Avg. Steps Needed||512→153|~70% reduction|
> |Total Wall-clock||41.8s→14.4s|~2.9× speedup|
> |
>
>
> LRD adds negligible FLOPs but incurs a 15% per-step latency increase (81.8ms→94.3ms) due to memory-bound operations (KL computation over 152k vocab). However, LRD reduces required steps by ~70% (512→153), yielding a net 2.9× end-to-end speedup.
>
> - Quality-vs-Time Pareto Frontier
>
> We plot a quality–time Pareto plot on HumanEval with Dream-Instruct-7B in Figure 8 (Appendix A.5 line 860), where we vary the diffusion step budget, measure absolute wall-clock time, and plot accuracy versus time for both baseline and LRD. Ours method demonstrates strict Pareto dominance: at any time budget, it achieves higher quality than baseline. At peak efficiency, LRD reaches 61.6% accuracy while being 3.7× faster than baseline's maximum of 56.1%.
>
>
> **5. Semi-AR Block**
> > For the semi-AR block setting, do beliefs transfer between blocks (e.g., previous block’s soft state informing the next), or is each block re-initialized?
>
> We thank the reviewer for this important clarification question.
>
> No, soft states (beliefs) do **not** transfer between blocks. Each block is initialised based on fully decoded tokens from previous blocks, not probabilistic distributions.  To clarify this explicitly, we provide a step-by-step semi-AR block decoding:
>
> `Step 1 - Initial Block Initialisation:`
> All positions in Block 1 are initialised as [MASK].
>
> `Step 2 - Initial Block Phase 1 (Latent Refinement):`
> Iteratively refine soft embeddings for Block 1 until convergence.
>
> `Step 3 - Initial Block Phase 2 (Predictive Feedback):`
> Progressively decode Block 1 until all positions are finalised as discrete tokens.
>
> `Step 4 - Subsequent Block Initialisation (t > 1):`
> Block t positions are initialised as [MASK], while all previous blocks (1,...,t-1) include completely decoded tokens only.
>
> `Step 5 - Subsequent Block Phase 1 (Latent Refinement):`
> Apply latent refinement within Block t only, conditioned on decoded tokens from previous blocks. No soft states from previous blocks.
>
> `Step 6 - Subsequent Block Phase 2 (Predictive Feedback):`
> Decode Block t until all positions are finalised as discrete tokens.
>
> `Step 7 - Repeat:`
> Repeat Steps 4-6 for each remaining block until all blocks are decoded.
>
> The key point is the **sequential decoding order** of blocks: Block $t$ is indeed conditioned on previous blocks, but when we begin decoding block $t$, all previous blocks ($1, ..., t-1$) have already been completely decoded. There are no remaining mask tokens in these blocks.

---

### Official Review · Reviewer_yc87 · 2025-11-01

**Soundness:** 4
**Presentation:** 4
**Contribution:** 4
**Rating:** 8
**Confidence:** 4

**Summary:**

This work introduces Latent Refinement Decoding (LRD), a two-stage framework that improves diffusion language model speed and generative quality on coding and math.

In Phase 1: Latent Refinement, [MASK] token embeddings are mixed with the embeddings of the top-p predicted tokens until distributional shift decreases below a threshold.

In Phase 2: Predictive Feedback Loop, standard low entropy denoising is applied to decode lowest entropy tokens, while mixing other tokens.

The Kullback-Liebler (KL) Divergence between iterative predictive steps is used to determine which tokens are finally decoded, enabling an early stopping mechanism for sampling.

Additional ablations are performed on variants of the decoding framework, revealing the importance of both stages of the framework as well as demonstrating the decoding speed improvement from early stopping.

**Strengths:**

- The presentation of this paper is very clear and easy to read
- The proposed method demonstrates improvements in both speed and generative performance on math and coding
- The figures give insight into method complexity and hyperparameter choices

**Weaknesses:**

- The evaluation is limited to two 7-8B parameter models and code and math, leaving models of different sizes and other natural language tasks unexamined
- The KL divergence is not compared to other statistical distances for monitoring saturation

**Questions:**

- Did you explore any other distances besides KL divergence?
- Instead of decoding the single lowest entropy token at each step, how is the performance of decoding $k$ tokens at a time in phase 2, such as if $i \in \textrm{top-k}(\\{H_t^{(j)}\\}_{j=1}^L)$?
- In Table 4 $E_\textrm{token}$, is the color-coding inverted? (Lower effective token count should be better/green)

---

> ### Author Response · Authors · 2025-11-23
> **Title:Response to Reviewer yc87 (Part 1/1)**
>
> We sincerely appreciate Reviewer yc87's constructive feedback and valuable suggestions. Our detailed responses are as follows.
>
> ---
>
> **1. Broader Evaluation Scope**
>
> > The evaluation is limited to two 7-8B parameter models and code and math, leaving models of different sizes and other natural language tasks unexamined
>
> To address this concern, we have expanded our evaluation to cover:
>
> - Different model sizes: We evaluate LRD on LLaDA2.0-mini-preview, a 16B MOE diffusion language model.
>
> |Model|Len|Method|HumanEval:Acc|HumanEval:Speed|MBPP:Acc|MBPP:Speed|GSM8K:Acc|GSM8K:Speed|MATH500:Acc|MATH500:Speed|
> |-|-|-|-|-|-|-|-|-|-|-|
> |LLaDA2.0-mini-preview|256|baseline|5.5|1.0×|19.2|1.0×|63.8|1.0×|16.2|1.0×|
> |||Ours|7.3(+1.8)|1.0×|22.2(+3.0)|1.1×|64.6(+0.8)|1.2×|17.0(+0.8)|1.0×|
> ||512|baseline|54.3|1.0×|54.7|1.0×|86.5|1.0×|44.6|1.0×|
> |||Ours|54.9(+0.6)|1.1×|58.2(+3.5)|1.6×|87.6(+1.1)|2.1×|45.8(+1.2)|1.1×|
> ||1024|baseline|74.2|1.0×|63.2|1.0×|87.7|1.0×|61.2|1.0×|
> |||Ours|78.7(+4.5)|1.1×|65.7(+2.5)|2.2×|88.9(+1.2)|3.8×|63.2(+2.0)|1.8×|
> |
>
> - Open-ended generation tasks beyond code/math: RACE [1] for reading comprehension and CNN/DailyMail [2] for summarisation.
>
> |Model|Len|Method|RACE:Acc|RACE:Speed|CNN/DM:BERTScore|CNN/DM:ROUGE-1|CNN/DM:ROUGE-L|CNN/DM:Speed|
> |-|-|-|-|-|-|-|-|-|
> |Dream-Ins-7B|256|baseline|81.1|1.0×|85.6|21.7|15.0|1.0×|
> |||Ours|82.2(+1.1)|1.3×|85.7(+0.1)|22.3(+0.6)|15.5(+0.5)|1.3×|
> ||512|baseline|81.9|1.0×|85.5|22.6|15.6|1.0×|
> |||Ours|82.4(+0.5)|1.4×|85.6(+0.1)|23.0(+0.4)|15.9(+0.3)|1.4×|
> ||1024|baseline|81.7|1.0×|85.7|22.8|15.7|1.0×|
> |||Ours|82.7(+1.0)|1.7×|85.8(+0.1)|23.3(+0.5)|16.5(+0.8)|1.7×|
> |LLaDA-Ins-8B|256|baseline|67.0|1.0×|82.3|19.9|13.9|1.0×|
> |||Ours|68.5(+1.5)|1.5×|82.3|20.0(+0.1)|13.9|1.7×|
> ||512|baseline|67.9|1.0×|82.2|19.8|13.7|1.0×|
> |||Ours|69.5(+1.6)|1.6×|82.4(+0.2)|20.1(+0.3)|13.9(+0.2)|1.8×|
> ||1024|baseline|68.6|1.0×|82.1|19.7|13.8|1.0×|
> |||Ours|70.6(+2.0)|1.7×|82.3(+0.2)|19.9(+0.2)|14.1(+0.3)|2.1×|
> |
>
> Results demonstrate that LRD's benefits generalise across: **Model scales**: Consistent improvements on different model sizes, with gains of +0.6 to +4.5 points across tasks. **Task types**: Improvements on reading comprehension (+0.5 to +2.0 on RACE) and summarisation (+0.1 to +0.8 on ROUGE scores), confirming LRD does not compromise generation quality on open-ended tasks. These experiments have been incorporated into the revised version, in the extended ablation study (Line 457) and Appendix A.2 (Line 762).
>
> [1] Lai et al. RACE: Large-scale ReAding Comprehension Dataset From Examinations. EMNLP 2017.
>
> [2] See et al. Get To The Point: Summarization with Pointer-Generator Networks. ACL 2017.
>
>
> **2. Other distances besides KL divergence**
>
> > The KL divergence is not compared to other statistical distances for monitoring saturation
>
> We have conducted a comparison of different distance metrics for monitoring convergence, shown in Figure 6 (Appendix A.3 line 784). KL divergence exhibits a clear decrease during both refinement and decoding, while alternative metrics (Wasserstein, Chi-Square, Euclidean, Cosimilarity) show high variance and noise, making them unreliable for detecting saturation.
>
> **3. Decoding Multiple Tokens Per Step**
>
> > Instead of decoding the single lowest entropy token at each step, how is the performance of decoding k  tokens at a time in phase 2
>
> Thank you for the suggestion. We ran an additional study where Phase 2 commits the top-$k$ lowest-entropy positions per step ($k \in {1,\dots,5}$). Across HumanEval, MBPP, GSM8K, and MATH500, accuracy drops sharply as $k$ increases while speedup grows; LRD consistently outperforms the hard-masking baseline for all $k$ in both accuracy and speed, but large $k$ yields an unfavorable quality–speed trade-off. This suggests that naïvely increasing $k$ is not an adequate way to introduce parallelism. Instead, following recent work such as Fast-dLLM, we also evaluate LRD combined with a confidence-aware parallel decoding scheme and observe higher speedups together with improved accuracy compared to Fast-dLLM alone. We have included the top-$k$ curves and the combined LRD+Fast-dLLM experiments in the extended ablation study (Line 457) and Appendix A.4 (Line 806).
>
> |top-k|Method|HumanEvalAcc|HumanEvalSpeed|MBPPAcc|MBPPSpeed|GSM8KAcc|GSM8KSpeed|MATH500Acc|MATH500Speed|
> |-|-|-|-|-|-|-|-|-|-|
> |1|baseline|50.6|1.0×|53.8|1.0×|75.3|1.0×|36.9|1.0×|
> |1|ours|56.9|1.2×|57.6|2.3×|78.2|1.8×|39.8|1.4×|
> |2|baseline|28.7|1.6×|39.6|1.8×|67.8|2.0×|28.2|1.9×|
> |2|ours|34.8|2.1×|42.2|4.5×|68.9|2.9×|30.2|2.8×|
> |3|baseline|22.6|2.2×|28.6|2.6×|54.2|2.9×|17.6|2.9×|
> |3|ours|26.8|2.9×|31.2|6.1×|57.2|4.4×|20.2|4.1×|
> |4|baseline|18.9|2.5×|22.4|3.2×|45.3|3.9×|9.0|3.7×|
> |4|ours|22.6|3.6×|24.6|7.7×|47.5|5.6×|11.4|5.3×|
> |5|baseline|8.5|2.8×|16.0|3.8×|29.2|4.8×|5.2|4.5×|
> |5|ours|11.6|4.2×|19.4|9.2×|31.5|6.7×|6.2|6.6×|
> |
>
>
>
>
> **4. Table 4 Color Coding**
>
> Thank you for catching this. We have corrected the colour coding in the revised version.

---

### Official Review · Reviewer_XKSR · 2025-11-01

**Soundness:** 3
**Presentation:** 3
**Contribution:** 3
**Rating:** 6
**Confidence:** 4

**Summary:**

The paper considers the diffusion-based LMs and proposes an approach to speed up dLLM decoding. The proposed Latent Refinement Decoding (LRD) approach aims to address two core limitations of previous dLLM decoding methods (e.g., LLaDa, Dream), namely, the information loss and the lack of well-behaved commitment dynamics. In particular, the method contains two stages, where the first stage maintains mixtures of predicted tokens and mask embedding and the second stage progressively finalizes confident tokens. Experimental results demonstrate the effectiveness and efficiency of the proposed LRD approach.

**Strengths:**

Overall, the paper is well-written. The material is presented in a clear and organized way. The method specifications, experimental results, ablation studies are informative. The method itself is well-motivated, with improvements in terms of effectiveness and efficiency over previous approaches.

**Weaknesses:**

The paper can further benefit from clarifications on following points (detailed in "Questions" section):

1. How does LRD prevent/avoid "mispredictions"?

1. What are hyper-parameter setups that recover previous methods with hard assignments?

1. (Minor issue) It might worth considering how to better phrase the second "core limitation."

**Questions:**

### 1. How does LRD prevent/avoid "mispredictions"?

LRD maintains the full density mixture over (top-p) token embedding and mask token embedding, and therefore, it is a not a hard masking/assignment. While it is very clear that hard assignment can potentially have an infinite $d_{KL}$ from the true posterior due to the lack of ability to correct for "misspecification" (line 48), it is not entirely clear to me how this soft assignment approach can prevent or avoid this issue. In other word, is there a guarantee (e.g., on the $d_{KL}$) on how close the convergence state is to the true posterior? I understand that a rigorous bound might be nontrivial to derive, but having some discussion (at least providing some intuition) could be very helpful.

### 1. What are hyper-parameter setups that recover previous methods with hard assignments?

Is there a way to use specific values of $(\alpha, \tau_{\text{refine}}, \tau_{\text{decode}})$ to recover previous hard masking/assignment approaches? I can imagine when $\alpha$ is set to 0, the mixture will be de-activated. Having this additional discussion (if feasible) can provide a clearer explanation of the relationship between LRD and previous approaches.

### 1. (Minor issue) It might worth considering how to better phrase the second "core limitation."

In Abstract, the second "core limitation" is "premature commitment." In Introduction, the second limitation is "inefficient convergence dynamics" which may involve premature (when being aggressive) or overdue (when being conservative) commitments. I am wondering if sth like the lack of well-behaved commitment dynamics would be a better fit.

---

> ### Author Response · Authors · 2025-11-23
> **Response to Reviewer XKSR (Part 1/1)**
>
> We thank Reviewer XKSR for the constructive comments and valuable suggestions. We address each point in detail below.
>
> ---
>
> **1. How does LRD prevent/avoid "mispredictions"?**
>
>
>
> We acknowledge that LRD cannot eliminate mispredictions at the final decoding stage. Our claim is that LRD reduces the probability of mispredictions by **delaying premature commitment**.
>
> **Intuition 1: Avoiding the "Infinite KL" Trap (Mathematical Perspective)**
>
> Hard assignment collapses distributions at each step: if an early step commits incorrectly, the probability of the true token becomes zero ($P(v^\*) \to 0$), theoretically yielding $\mathrm{KL}(q^* \\| q_{\mathrm{hard}}) \to \infty$ with no mechanism for recovery.
>
> In contrast, LRD maintains a non-zero probability mass over tokens throughout the latent refinement phase. Therefore, if we assume that **the true token $v^*$ is included within the top-p candidates** with high probability, the assigned probability mass in denoising is strictly bounded away from zero (specifically, by $\alpha_t \cdot \hat{p}(v^*)$). Consequently, the KL divergence is **upper-bounded by a finite value**, avoiding the singularity seen in hard assignment.
>
> Intuitively, this bound takes the form:
>
> $KL(q^\*\|q_{\text{soft}}) \leq -\log(\alpha_t \cdot \hat{p}(v^\*)) = \log\frac{1}{\alpha_t \cdot \hat{p}(v^\*)} < \infty$
>
> with high probability.
>
> This finite bound ensures that the update signals remain informative, allowing the model to recover from early uncertainties rather than getting stuck in a zero-probability trap. This theoretical intuition explains the **monotonic KL decrease** observed in Figure 2 (Line 335), showing that refinement consistently improves distribution quality.
>
> **Intuition 2: Linear Propagation of Uncertainty (Mechanism Perspective)**
>
> Another intuition is that LRD encodes distributional information through mixed embeddings at each position. Since the transformer's self-attention mechanism is linear with respect to embeddings, these mixed representations allow the model to propagate uncertainty information across positions. This enables the model to iteratively refine token hypotheses using **global context** to correct early local uncertainties. Our core assumption is that distributions refined with global context are more accurate than premature local predictions, an intuition empirically validated by the consistent accuracy gains across models and datasets shown in Table 1 (Line 270).
>
>
> **2. Recovering previous hard-assignment methods**
>
> > What are hyper-parameter setups that recover previous methods with hard assignments?
>
>
> The hard-masking decoders in Dream/LLaDA can be recovered as special cases of our framework via the following hyper-parameter choices:
>
> - **Mixing coefficient:** set the maximum mix ratio to zero ($r_f = 0$, hence $\alpha_t^{(i)} = 0$), so undecided positions always use the plain $[\text{MASK}]$ embedding and decoding reduces to standard hard masking.
> - **Refinement steps:** set $T_{\text{refine}} = 0$ to skip the refinement-thinking phase and start decoding directly from a fully masked state, as in prior work.
> - **Early stopping:** disable KL-based early stopping ($\tau_{\text{decode}} = 0$), matching the fixed-step decoding used in Dream/LLaDA.
>
> In summary, under $(r_f = 0, T_{\text{refine}} = 0, \tau_{\text{decode}} = 0)$, our sampler is equivalent to the original hard-assignment baselines.
> We have made this mapping explicit in Appendix A.1 (line 756) of the revised paper.
>
>
>
>
> **3. Terminology Consistency**
>
> Thank you for pointing out this inconsistency. We agree that the original wording was confusing. In the revised version, we have unified the description of the second core limitation in both the abstract and the introduction as a **lack of well-behaved commitment dynamics**, following your suggestion. This makes the high-level problem statement conceptually consistent across sections.

---

### Note · Authors · 2026-02-19

**Comment:**

We are withdrawing this submission but leaving this clarification to address significant factual inaccuracies in the Meta Review. Given our consistently positive initial scores (**8/6/6/6; confidence 4/4/3/3**) and the suspension of the discussion phase due to the ICLR reviewer leak, a strictly factual Meta Review was critical.

We correct the record on the following points:

- **Misattributed claims:** The Meta Review cites a lack of evidence for an “error correction” hypothesis as a primary weakness. This was never our claim; it was a “future benefit” independently suggested by Reviewer XKSR. It is procedurally flawed to penalize our manuscript for a reviewer’s hypothetical framing.

- **Use of terminology not present in our paper:** The AC critiques our method using terms like “heuristic interacts,” “concrete score,” and “time-evolution.” None of these terms appear anywhere in our paper.

- **False claims of undefined notation:** The AC claims key mathematical notation is undefined. Factually, $H_t^{(j)}$ is explicitly defined alongside Equation 2, and $p_t^{(i)}(v)$ is defined alongside Equation 4.

- **Ignored rebuttal & evaluation scope:** The claim that we evaluated “only two base models” is factually incorrect and ignores our rebuttal. **LLaDA and Dream are currently the de facto standard open-source discrete diffusion LLMs.** The AC appears to have overlooked our revised manuscript, which demonstrated consistent gains across **6 model variants** (including the 16B LLaDA2.0-mini-preview) and **6 distinct tasks** (including open-ended generation such as RACE and CNN/DailyMail), alongside extensive Fast-dLLM and KV-cache integration studies.

We thank the reviewers for their time, rigorous engagement, and positive recognition of our work.

**Withdrawal Confirmation:**

I have read and agree with the venue's withdrawal policy on behalf of myself and my co-authors.

---

### Meta-Review · Area_Chair_8ngB · 2025-12-26

**Summary:**

The paper considers a two-phase decoding strategy for diffusion-based LMs, in which Phase 1 evolves the (continuous, soft) embeddings of the tokens, and Phase 2 "snaps" them into discrete decisions. They operationalize Phase 1 by evolving them with the outputs of the trained model, and tracking convergence using KL divergence as a metric. They operationalize Phase 2 by unmasking the lowest-entropy tokens, and again, tracking convergence to possibly early-stop using KL divergence.

On the positive side, the paper proposes a training-free method that improves baseline (decoding) performance, largely without sacrificing latency/efficiency, on two base models LLaD and Dream. During the discussion, the authors provided some evidence that their proposed strategy can be combined with some more modern decoding strategies like Fast-dLLM and/or KV-Cache optimization strategies.

On the negative side, as reviewer XKSR notes, the heuristic is motivated by a hypothesis of some failure modes of standard decoding --- but it's unclear what support there is for this hypothesis. Indeed, there aren't even examples / anecdotal studies in the paper. The authors do provide some intuition in a response, and in some additional prose and calculation added in the appendix --- but I think this still doesn't get to the core of the issue. Additionally, I do not feel like the authors did an adequate job of explaining how their heuristic interacts with the super-standard view of decoding strategies as discretizations of a continuous reverse diffusion process --- and in particular, whether their heuristic could actually *introduce* errors. In particular, it isn't clear how to think of the embedding evolution process in tandem with the time-evolution of the reverse diffusion, or the concrete score, or anything in terms of the "standard" interpretation of discrete diffusion models.

I also think that the mathematical/technical writing in the paper is very hard to track. For example, equations (3)-(4) reference p_t^{i}(v), which isn't defined until Line 212. Even there, one is only left to impute what dLLM_theta^i is --- it's never defined. In equation (7), H_t^{j} is used but never defined, again --- is it the j-th hidden layer? Why this choice? These are not minor things --- they are essential to the mathematical definition of the method.

Finally, as reviewer b4sQ points out, the method's integration is only shown with two based models --- LLaDa and Dream --- and it isn't clear how consistent the gains would be across different models. Similarly, as several reviewers note, on the benchmarks considered like GSM8K and MATH500, the baseline models are already quite good --- so the small-to-moderate gains in accuracy are of unclear value beyond these benchmarks.

**Reviewer Concerns:**

Limited base model consideration, gains are often modest and on benchmarks on which baseline models already perform well, technical writing is sloppy and hard to follow, unclear when the method helps / harms. There were rebuttals by the authors on all these points, and I think the authors made some good points, but I don't think the concerns were fully addressed.

**Reviewer Scores:**

Reviewers didn't follow up to the rebuttal of the authors unfortunately, so I can only say that for myself --- I think the authors made some good points, but the concerns weren't fully addressed.

---

### Decision · Program_Chairs · 2026-01-26

Reject